# Whole-organism behavioral profiling reveals a role for dopamine in state-dependent motor program coupling in *C. elegans*

Nathan Cermak[†], Stephanie K Yu[†], Rebekah Clark[†], Yung-Chi Huang, Saba N Baskoylu, Steven W Flavell*

Picower Institute for Learning & Memory, Department of Brain & Cognitive Sciences, Massachusetts Institute of Technology, Cambridge, United States

**Abstract** Animal behaviors are commonly organized into long-lasting states that coordinately impact the generation of diverse motor outputs such as feeding, locomotion, and grooming. However, the neural mechanisms that coordinate these distinct motor programs remain poorly understood. Here, we examine how the distinct motor programs of the nematode *C. elegans* are coupled together across behavioral states. We describe a new imaging platform that permits automated, simultaneous quantification of each of the main *C. elegans* motor programs over hours or days. Analysis of these whole-organism behavioral profiles shows that the motor programs coordinately change as animals switch behavioral states. Utilizing genetics, optogenetics, and calcium imaging, we identify a new role for dopamine in coupling locomotion and egg-laying together across states. These results provide new insights into how the diverse motor programs throughout an organism are coordinated and suggest that neuromodulators like dopamine can couple motor circuits together in a state-dependent manner.

*For correspondence:
flavell@mit.edu

[†]These authors contributed equally to this work

**Competing interests:** The authors declare that no competing interests exist.

## Introduction

As animals explore their environments, their nervous systems transition between a wide range of internal states that influence how sensory information is processed and how behaviors are generated (*Anderson and Adolphs, 2014*; *Artiushin and Sehgal, 2017*; *Liu and Dan, 2019*). These internal states of arousal, motivation, and mood typically alter many ongoing behaviors, impacting motor circuits that control diverse behavioral outputs such as feeding, grooming, and locomotion. A full understanding of how internal states are generated should explain how they coordinate the production of many complex motor outputs.

Neuromodulators play a central role in generating internal states. For example, neuropeptides like orexin and pigment dispersing factor (PDF) promote brain-wide states of wakefulness in mammals and flies, respectively (*Saper et al., 2010*; *Taghert and Nitabach, 2012*). In awake animals, norepinephrine controls arousal and attention (*Aston-Jones and Cohen, 2005*; *Carter et al., 2010*), the neuropeptides Tac2 and CRH induce states of heightened anxiety (*Füzesi et al., 2016*; *Kormos and Gaszner, 2013*; *Zelikowsky et al., 2018*), and the neuropeptides AgRP and NPY promote behaviors associated with hunger (*Chen et al., 2019*; *Krashes et al., 2013*). The anatomical organization of neuromodulatory systems makes them well-suited to impact a wide range of motor outputs: the projections of neuromodulator-producing neurons are typically very diffuse, targeting many brain regions. However, our mechanistic understanding of how neuromodulatory regulation of CNS circuits is propagated to diverse motor outputs remains limited.

**eLife digest** Animals generate many different motor programs (such as moving, feeding and grooming) that they can alter in response to internal needs and environmental cues. These motor programs are controlled by dedicated brain circuits that act on specific muscle groups.

However, little is known about how organisms coordinate these different motor programs to ensure that their resulting behavior is coherent and appropriate to the situation. This is difficult to investigate in large organisms with complex nervous systems, but with 302 brain cells that control 143 muscle cells, the small worm *Caenorhabditis elegans* provides a good system to examine this question.

Here, Cermak, Yu, Clark et al. devised imaging methods to record each type of motor program in *C. elegans* worms over long time periods, while also dissecting the underlying neural mechanisms that coordinate these motor programs. This constitutes one of the first efforts to capture and quantify all the behavioral outputs of an entire organism at once.

The experiments also showed that dopamine – a messenger molecule in the brain – links the neural circuits that control two motor programs: movement and egg-laying. A specific type of high-speed movement activates brain cells that release dopamine, which then transmits this information to the egg-laying circuit. This means that worms lay most of their eggs whilst traveling at high speed through a food source, so that their progeny can be distributed across a nutritive environment.

This work opens up the possibility to study how behaviors are coordinated at the level of the whole organism – a departure from the traditional way of focusing on how specific neural circuits generate specific behaviors. Ultimately, it will also be interesting to look at the role of dopamine in behavior coordination in a wide range of animals.

In the simple nematode *C. elegans*, it should be feasible to determine how internal states influence every behavioral output of the animal. The *C. elegans* nervous system consists of 302 neurons, which act on 143 muscle cells (*White et al., 1986*). *C. elegans* generates a well-defined repertoire of motor programs, each with a devoted motor circuit and muscle group: locomotion, egg-laying, feeding, defecation, and postural changes of the head and body (*de Bono and Maricq, 2005*; *Collins et al., 2016*; *Pirri et al., 2009*; *Schafer, 2005*; *Stephens et al., 2008*). Quantitative studies of *C. elegans* locomotion have shown that animals switch between long-lasting behavioral states. In an extreme case, *C. elegans* animals cease all behaviors as they enter sleep-like states during development and after periods of stress (*Nichols et al., 2017*; *Raizen et al., 2008*; *Van Buskirk and Sternberg, 2007*). Awake animals in a food-rich environment switch between roaming and dwelling states, where they either rapidly explore the environment or restrict their movement to a small area (*Ben Arous et al., 2009*; *Flavell et al., 2013*; *Fujiwara et al., 2002*; *Stern et al., 2017*). After removal from food, *C. elegans* generates an area-restricted search state before switching to a dispersal state (*Gray et al., 2005*; *Hills et al., 2004*; *López-Cruz et al., 2019*; *Wakabayashi et al., 2004*). Like the internal states of mammals, these behavioral states in *C. elegans* last from minutes to hours and the transitions between states are abrupt. It remains unclear how the full repertoire of *C. elegans* behaviors is coordinated as awake *C. elegans* animals switch between behavioral states.

Neuromodulation in *C. elegans* plays a pivotal role in behavioral state control (*Chase and Koelle, 2007*; *Li and Kim, 2008*). Serotonin initiates and maintains dwelling states, while the neuropeptide PDF initiates and maintains roaming states (*Choi et al., 2013*; *Flavell et al., 2013*; *Horvitz et al., 1982*; *Rhoades et al., 2019*; *Sawin et al., 2000*; *Stern et al., 2017*). In addition, dopamine, tyramine, and octopamine influence behaviors associated with the presence or absence of food (*Alkema et al., 2005*; *Chase et al., 2004*; *Horvitz et al., 1982*; *Sawin et al., 2000*; *Stern et al., 2017*). The *C. elegans* sleep-like state is also strongly influenced by neuropeptides, including NLP-8, FLP-13, and FLP-24, and ligands for the NPR-1 receptor (*Choi et al., 2013*; *Iannacone et al., 2017*; *Nath et al., 2016*; *Nelson et al., 2014*). Command-like neurons that release neuromodulators are capable of driving state changes: the serotonergic NSM neurons induce dwelling states (*Flavell et al., 2013*; *Rhoades et al., 2019*) and the peptidergic ALA neuron evokes a sleep-like state (*Hill et al., 2014*; *Nath et al., 2016*; *Nelson et al., 2014*). However, there is not a one-to-one mapping between neuromodulators and specific states. For example, although serotonin release

from NSM drives dwelling, serotonin release from ADF neurons has no apparent effect on dwelling and instead impacts other behaviors (*Flavell et al., 2013*). Likewise, ALA releases at least three neuropeptides that have unique yet overlapping functions to inhibit downstream behaviors during sleep (*Nath et al., 2016*). However, each of these neuropeptides is also produced by other neurons that do not induce sleep. Thus, individual neuromodulators may exert different effects during different states and their combinatorial actions might give rise to the widespread behavioral changes that accompany each state.

In this study, we examine how neuromodulation coordinates diverse behavioral outputs to give rise to state-dependent changes in behavior. First, we describe a new imaging platform that permits simultaneous, automated quantification of each of the main *C. elegans* motor programs. Then we analyze hours-long behavioral recordings to fully characterize the behavioral states of this animal. Finally, we identify a dopaminergic pathway that couples multiple motor programs together as animals switch behavioral states, revealing that dopamine promotes egg-laying in a locomotion state-dependent manner. Our study provides new insights into how the diverse motor programs throughout an organism are coordinated and suggests that neuromodulators like dopamine can couple motor circuits together in a state-dependent manner.

## Results

### The distinct motor programs in *C. elegans* are coordinated with one another

To examine how the distinct *C. elegans* motor programs are coordinated, we designed and constructed tracking microscopes and an accompanying software suite that permits simultaneous, automated measurement of numerous behaviors (*Figure 1A*; *Figure 1—figure supplement 1*). The microscope, which shares several design features with previously described microscopy platforms (*Yemini et al., 2013*; see also *Faumont and Lockery, 2006*; *Nguyen et al., 2016*; *Venkatachalam et al., 2016*), collects brightfield images of individual animals (*Figure 1B*) at a frequency of 20 Hz and resolution of 1.4 um/pixel, which is sufficient to capture the most rapid and small-scale movements of *C. elegans*, such as pharyngeal motion (*Avery and You, 2012*; *Lockery et al., 2012*). A closed-loop tracking system reliably keeps animals in view over hours or days, while live data compression makes storage of approximately 1.7 million images per animal per day feasible. Although the parameters for data collection that we use here are tailored to *C. elegans* recordings, this low-cost, open-source microscopy platform should be useful for recordings of many small animals (see Materials and methods for links to parts list and build tutorial).

We developed a software suite that allows us to automatically extract measurements of each of the main *C. elegans* motor programs from these video recordings. Because *C. elegans* is transparent, each animal movement – including movements inside the body, like those of the pharynx – is visible during brightfield imaging. Using this software suite, we are able to extract: body/head posture, locomotion, egg-laying, defecation, and pharyngeal pumping (i.e. feeding). Extracting body posture and locomotion was straightforward, except during omega bends (see Materials and methods) (*Stephens et al., 2008*; *Figure 1B*; *Figure 1—figure supplement 2A–D*), However, the detection of egg-laying, defecation, and pumping required us to implement tailored machine vision algorithms (*Figure 1—figure supplement 2E–O*; see Materials and methods). To ensure the reliability of our measurements, we compared the automated scoring of egg-laying, defecation, and pumping to manually scored data and found a high level of concurrence (*Figure 1—figure supplement 2G,J,O*). These advances now allow us to extract near-comprehensive records of each animal's behavior from hours-long recordings. The resulting datasets capture behavioral outputs over multiple timescales: from milliseconds-scale postural changes to hours-long behavioral states (*Figure 1C–E* shows an example dataset at multiple time resolutions).

To examine how the distinct motor programs of *C. elegans* are coordinated over time, we first analyzed the relationships between each behavioral variable. We examined data from 30 adult well-fed wild-type animals, each recorded for six hours on a homogenous *E. coli* food source. To determine how each behavioral output varies as a function of locomotion, we examined average behavioral outputs across different animal velocities (*Figure 1F*). Egg-laying events were predominant during rapid forward locomotion, while defecation events were most commonly observed during

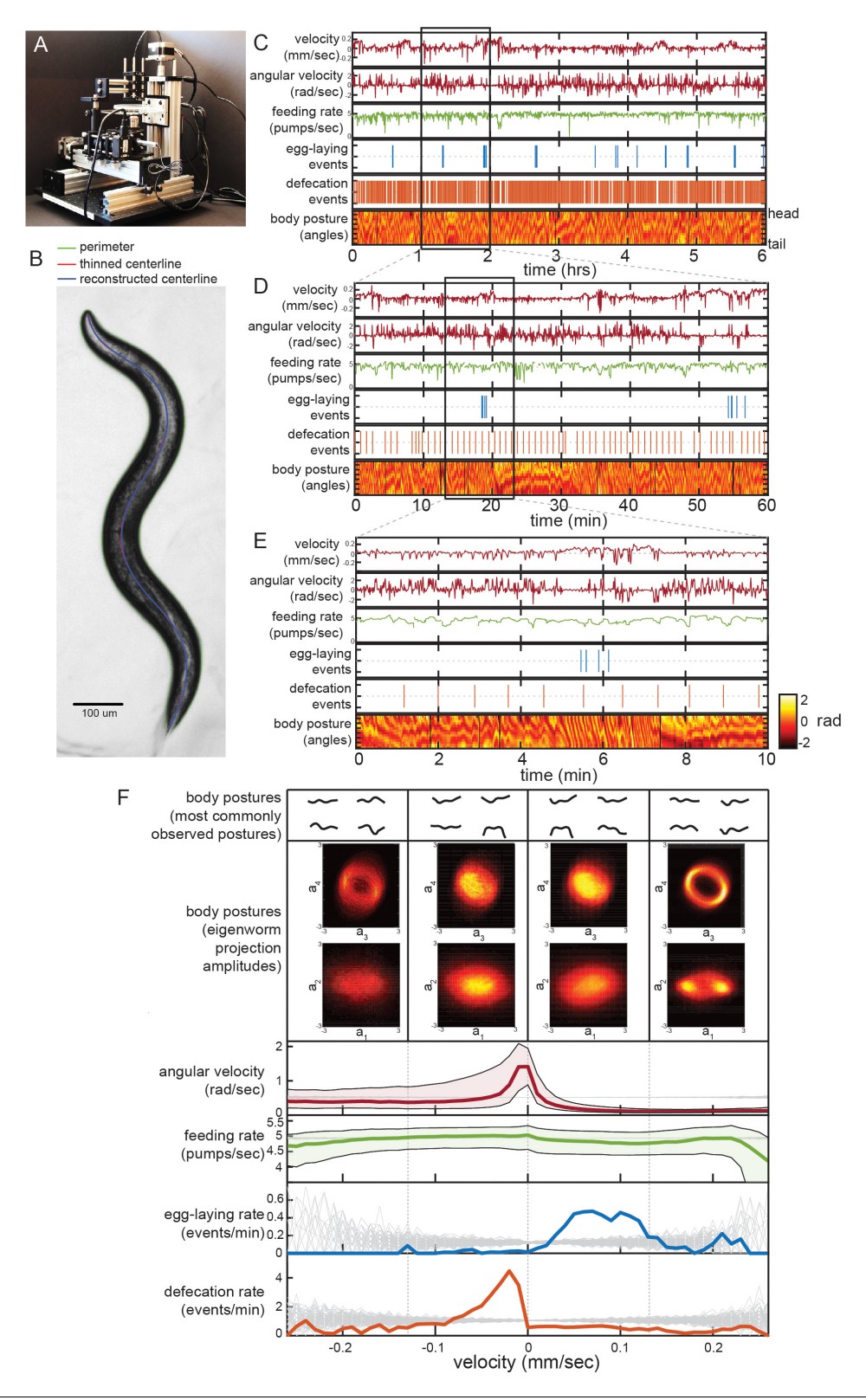

**Figure 1.** Simultaneous measurement of the diverse *C. elegans* motor programs. (**A**) Image of the tracking microscope. (**B**) Example image of a *C. elegans* animal from the tracking microscope. Green line denotes detected outline of worm; red line indicates worm's centerline, obtained by thinning the thresholded image; blue line is the centerline as reconstructed from a spline-based 14-parameter representation. (**C–E**) Example dataset from the tracking microscope, showing the main *C. elegans* motor programs over 6 hr (**C**), 1 hr (**D**), or 10 min (**E**). (**F**) Average behaviors observed while animals

*Figure 1 continued on next page*

*Figure 1 continued*

travel at different velocities. Data are from 30 wild-type animals, and data points were separated into bins based on instantaneous velocity. The average motor outputs in each bin are plotted. For angular velocity and feeding rate, data are shown as medians, as well as 25th and 75th percentiles. For egg-laying rate and defecation rate, data are shown as mean event rates (gray lines indicate random samples). For posture, data were segregated into four bins ranging from −0.25 mm/sec to 0.25 mm/sec. For each bin, we analyzed posture in two ways: by identifying and displaying the four most commonly observed postures of the 100 compendium postures described in *Figure 2* (top), or by plotting 2D histograms of the projection amplitudes of the four eigenworms (middle). Eigenworm analysis was conducted using the full datasets using previously described methods (*Stephens et al., 2008*; and see Materials and methods) and then the projection amplitudes in each velocity bin were plotted as 2D histograms.

The online version of this article includes the following figure supplement(s) for figure 1:

**Figure supplement 1.** Schematic of the tracking microscope.
**Figure supplement 2.** Extraction of behavioral parameters from video recordings, and validation of methods.
**Figure supplement 3.** Event-triggered averages showing behavioral coordination surrounding egg-laying and DMP events.

slow or reverse movement. We also examined typical body postures as a function of velocity in two ways. First, we identified the most commonly observed body postures at different velocities (top of *Figure 1F* shows most common postures out of 100 reference postures; see below). Second, we plotted 2D histograms of the eigenmodes of the angles along the body when animals traveled at different velocities (*Figure 1F*, middle) (*Stephens et al., 2008*). Differences in the amplitudes of the eigenmodes correspond to differences in body postures. From these histograms and from the set of most common body postures, it is evident that the postures that animals express co-vary considerably with velocity, suggesting that animals sample different body postures at different velocities (*Figure 1F*). We also examined average behaviors that surround egg-laying and defecation events by plotting event-triggered averages (*Figure 1—figure supplement 3*). These analyses revealed many of the same relationships between the behavioral outputs, but also showed that egg-laying and defecation events each occur during stereotyped posture and locomotion changes, as has been previously reported (*Collins et al., 2016*; *Hardaker et al., 2001*; *Nagy et al., 2015*). We conclude from these analyses that there is extensive coordination between the distinct *C. elegans* motor programs.

## Unsupervised classification of *C. elegans* behavioral states based on postural changes

To understand how these behaviors are coupled together over longer time scales, we sought to identify the long-lasting behavioral states that wild-type animals generate, so that we could determine which motor outputs are observed in each state. Unsupervised learning can reveal the underlying states that generate observed behavioral variables (*Berman et al., 2014*; *Marques et al., 2018*; *Wiltschko et al., 2015*). We and others have previously applied hidden Markov models (HMMs) to *C. elegans* locomotion parameters, which identifies roaming and dwelling states, as well as quiescence/sleep under certain conditions (*Flavell et al., 2013*; *Gallagher et al., 2013*). However, there may be a broader set of behavioral states that differ along other behavioral axes. To identify such states, we performed unsupervised discovery on the animal's body posture (*Figure 2A*), which is the most complex behavioral variable and is coordinated with the other motor programs.

To describe postural changes in a compact manner, we used a previously described approach where we learned a compendium of reference postures that encompasses the broad range of body postures that animals display (*Figure 2A–B*; *Schwarz et al., 2015*). To describe posture at any time point, we match the actual posture of an animal to its most similar match in the compendium. We then constructed a transition matrix that describes the probability of switching from each posture to the others (*Figure 2C*). Clustering the rows of this matrix revealed a striking, symmetric block-like structure, indicating that there are groups of postures, which we term 'posture groups,' that animals transition back and forth between (*Figure 2C–D*; 11.9-fold higher transition rate to postures within the same group). The postures within individual groups included those emitted during both forward and reverse movement (*Figure 2—figure supplement 1*), indicating that the time spent in a given posture group can span multiple forward-backward transitions. This organization suggests some degree of long-term stereotypy in how animals emit their postures while exploring a bacterial food lawn.

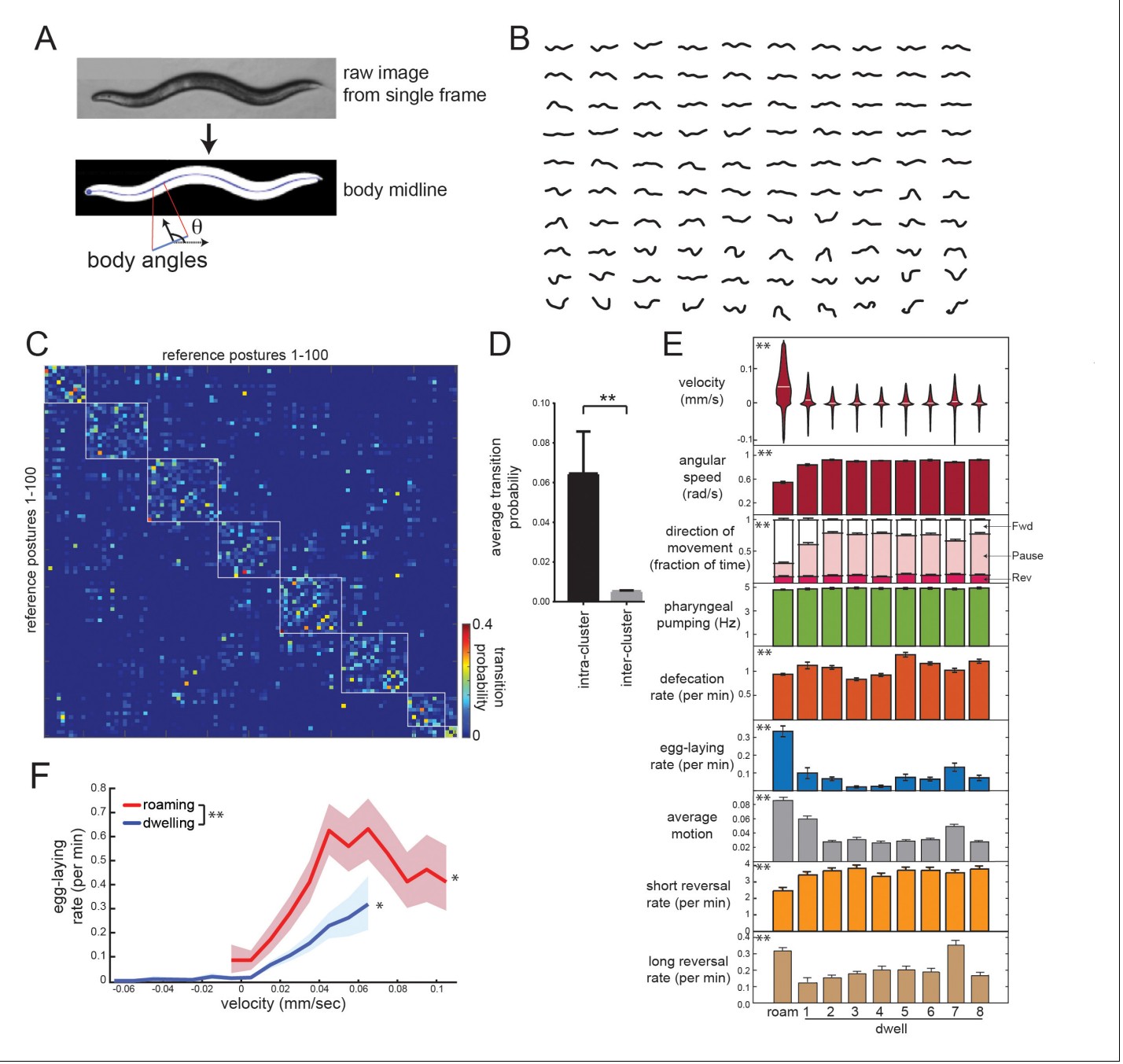

**Figure 2.** Identifying behavioral states through time series analysis of *C. elegans* posture. (A) Schematic showing that body posture is quantified as a vector of relative body angles, from head to tail. (B) Compendium of 100 reference postures that encompass the range of typical *C. elegans* postures. The compendium was derived by hierarchical clustering of many observed postures. (C) Transition matrix that shows probability of transitioning from each reference posture to the others. Self-transitions were excluded from this analysis. The rows of the matrix were clustered, and reference postures are sorted according to their cluster membership (eight clusters total; white boxes). (D) Average transition rates between postures within the same cluster, versus average transition rates between postures in different clusters. **p<0.01, Wilcoxon signed rank test. (E) Average behaviors in each hidden state. Velocity is shown as a violin plot, while other behaviors are shown as averages across 30 wild-type animals. Data are means ± standard error of the mean (SEM). 'Average Motion' here is defined as the average standard deviation of each of the 14 body angles over 2 s intervals throughout the state, which measures the degree to which body angles change over rapid timescales. Short reversals were backwards movements that lasted <4 s; long reversals lasted >4 s. **p<0.001, behaviors vary across states, Friedman test. (F) Average egg-laying frequency as a function of animal velocity, during roaming and dwelling states. Within each state, data points were segregated into bins based on ongoing animal velocity and egg-laying frequency was calculated across all the data points in each bin. Bins with <4 min of data per animal were excluded due to an insufficient quantity

*Figure 2 continued*

of data to warrant analysis. n = 30 animals. Error shading indicates 95% confidence intervals. **p<0.05, roaming vs dwelling, Bonferroni-corrected permutation test. *p<0.05, positive correlation of velocity versus egg-laying, empirical bootstrap test.

The online version of this article includes the following figure supplement(s) for figure 2:

**Figure supplement 1.** Locomotion during the 100 compendium postures.
**Figure supplement 2.** Additional analyses related to Posture-HMM.
**Figure supplement 3.** Further characterization of behavioral states from Posture-HMM.
**Figure supplement 4.** Analysis of the dwelling sub-modes.
**Figure supplement 5.** The posture-HMM generates posture sequences that resemble those from real animals.
**Figure supplement 6.** Locomotion surrounding egg-laying events.
**Figure supplement 7.** Stereotyped behavioral changes accompany state transitions.

To more precisely characterize this long-term stereotypy, we fit an HMM where the emissions are essentially these posture groups (see Materials and methods; *Figure 2—figure supplement 2A–C*). Based on Bayesian information criteria (BIC) estimates, a model with nine hidden states provided the best fit to the recorded data (*Figure 2—figure supplement 2D*). Importantly, we reliably converged to the same model parameters, even when training was performed on different sets of animals and from different random starting conditions (*Figure 2—figure supplement 2E–F*). These results suggest that the posture-HMM can provide a reliable description of postural changes over time.

## Posture-HMM identifies the roaming state and distinct sub-modes of dwelling

To understand the behavioral states that were captured by the HMM (*Figure 2—figure supplement 3A*), we first examined locomotion parameters in each of the states. One of these behavioral states consisted of high forward velocity, low angular speed and typically lasted from tens of seconds to many minutes (*Figure 2E*). Based on these parameters, this state is equivalent to the previously defined roaming state (*Figure 2E*; labeled 'Roam') (*Ben Arous et al., 2009*; *Flavell et al., 2013*; *Fujiwara et al., 2002*). Animals traveled at lower velocities during the other eight states, but, in contrast to quiescence/sleep states, they maintained high pumping and defecation rates in each (*Figure 2E*). To maintain consistency with previous literature, we describe these as dwelling sub-modes (Dwell1-8; see *Video 1* for examples). Quiescence states were not recovered from this analysis because this state was almost never observed under our recording conditions (only five ~20 s bouts identified in ~180 hr of data; *Figure 2—figure supplement 3B*).

Because these dwelling sub-modes had not been previously described, we characterized them further. Whereas five of these states (Dwell2,3,4,5,8) reflected almost completely paused movement, the other three states consisted of characteristic movements: animals displayed steady, slow forward locomotion in Dwell1 (*Figure 2E*; see movement direction and average motion), a high incidence of active reversing in Dwell7 (*Figure 2E*; see average motion and long reversal rates), and a wider range of head and neck movements in Dwell6 (*Figure 2—figure supplement 3A*; see standard deviation of angles along body); Moreover, animals displayed stereotyped body postures in each of the dwelling sub-modes that were significantly different from one another (*Figure 2—figure supplement 3A*; *Video 1*). For example, animals in Dwell4 displayed a reliably flat body posture with only very shallow bends (*Figure 2—figure supplement 3A*; see most common postures). The average duration of each sub-mode was ~10 s (*Figure 2—figure supplement 4A–B*) and animals transitioned between them in a non-random fashion: the transition rates between many sub-modes were close to zero, whereas other transition rates were quite high (*Figure 2—*

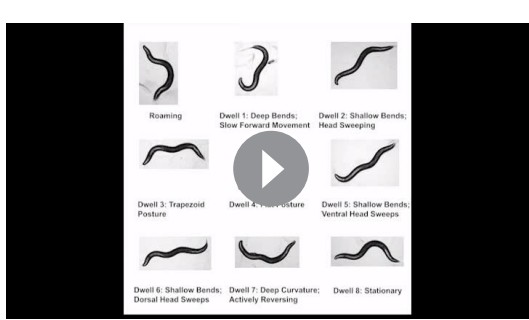

**Video 1.** Examples of behavioral states captured through posture-HMM. This video shows examples of the nine different behavioral states identified through posture-HMM. Note that videos are of different durations and are looped.
https://elifesciences.org/articles/57093#video1

*figure supplement 4C–D*). This suggests that animals transition between these distinct sub-modes of dwelling in an organized fashion.

We considered whether segmenting dwelling into these discrete sub-modes provides a better description of postural changes during dwelling than simpler models where animals continuously switch between postures without any further temporal structure. To investigate this, we asked whether posture sequences generated by the posture-HMM provided a better match to posture sequences from real animals, compared to simpler alternatives. We compared the posture-HMM to two alternative models: (1) a model involving continuous transitions between postures, where the probability of each posture transition was determined by our measurements from real animals (*Figure 2C*); and (2) a model involving continuous transitions between postures, where the probability of each posture transition was determined by the similarity between postures. We generated synthetic posture sequences from each model and quantified how well they matched the sequences from real animals by measuring the average duration of time between each of the 100 compendium postures in the synthetic and real animal data (*Figure 2—figure supplement 5A*). This metric captures the higher order patterns in which animals emit their postures, but does not assume any particular underlying structure. Based on this criterion, the posture sequences generated by the posture-HMM were a significantly better match to the sequences from actual animals, compared to the alternative models (*Figure 2—figure supplement 5B*). These results, together with the BIC analysis described above (*Figure 2—figure supplement 2D*), suggest that the posture-HMM provides a robust description of how animals transition between distinct postures during dwelling.

## Coordinated changes in multiple motor programs during behavioral states and state transitions

Having examined how posture and locomotion differ across these distinct states, we next quantified the occurrence of other motor programs in roaming states and each dwelling sub-mode (*Figure 2E*). Egg-laying rates were highest during roaming, whereas the dwelling sub-modes had smaller, but significant, differences in their egg-laying rates. This is consistent with previous observations that egg-laying is often accompanied by increased movement (*Hardaker et al., 2001*; *McCloskey et al., 2017*). Previous work showed that animals display an acute reduction in speed immediately after egg-laying (*Hardaker et al., 2001*). We found that this rapid speed change is present after egg-laying events during roaming and dwelling, but is less pronounced when eggs are laid during roaming (*Figure 2—figure supplement 6*). Defecation rates were also significantly different between the nine states, but feeding rates were the same across the states (*Figure 2E*). Taken together, these data suggest that animals transition between a high-velocity/high-egg-laying roaming state and different sub-modes of dwelling in which they display distinct body postures and motor programs.

Animals also displayed stereotyped changes in multiple motor programs at the moments when they switched between behavioral states (*Figure 2—figure supplement 7*). We characterized these effects by examining event-triggered averages timed to each type of behavioral state transition. We found that the state transitions were accompanied by transient and reliable changes in velocity, pumping, and defecation rates (*Figure 2—figure supplement 7*). For example, transitions from Dwell2 to Dwell4 were preceded by a high incidence of defecation motor program (DMP) events (immediately before the transition), followed by a transient increase in velocity and a reduction in feeding. These observations further confirm that the posture-HMM identified state transitions that correspond to reliable changepoints in behavior and suggest that animals display reliable, multimodal behavioral trajectories as they switch between these stereotyped postural states.

## Dopamine signaling promotes egg-laying during roaming states

We next sought to identify and characterize the neural circuits that allow animals to coordinate their motor programs across behavioral states. Here, we focused on the coupling between the roaming state and egg-laying, since this was the most robust form of motor program coupling that we detected and it could be easily measured using our new microscopy platform. We considered whether the increased frequency of egg-laying during roaming could be fully explained by animals laying eggs more frequently when they move faster. Thus, we examined egg-laying frequency as a function of velocity within roaming and within dwelling. While there was a positive correlation between velocity and egg-laying rate within each state, we observed a universally higher egg-laying

rate during roaming compared to dwelling, even when matched for velocity (*Figure 2F*). These results suggest that animals lay more eggs while traveling at high speeds, particularly during high-speed roaming states.

To test whether neuromodulation is critical for the coupling between locomotion state and egg-laying, we examined mutants lacking biogenic amine and neuropeptide neuromodulators. For each mutant, we characterized the time spent in each behavioral state and the motor outputs within each state (*Figure 3A*). States were defined consistently across genotypes using parameters learned from wild-type animals. Serotonin-deficient *tph-1* mutants displayed reduced dwelling, higher speeds during roaming, reduced feeding, and fewer egg-laying events, as has been previously described (*Figure 3A–B*; *Avery and You, 2012*; *Flavell et al., 2013*; *Hobson et al., 2006*; *Horvitz et al., 1982*). However, the increase in egg-laying rates during roaming states was largely intact, suggesting that serotonin is dispensable for this form of motor program coupling. *pdfr-1* mutants, deficient in PDF neuropeptide signaling, displayed a broad defect in roaming states: they spent less time roaming, traveled at lower velocity while roaming, and did not bias their egg-laying to the state as robustly as wild-type animals. Thus, although *pdfr-1* mutants are defective in locomotion and egg-laying during roaming, this may be part of a general deficit in their ability to display roaming states (*Flavell et al., 2013*), rather than a specific deficit in motor program coupling. *tdc-1* mutants that are defective in tyramine and octopamine displayed increased time in the roaming state. However, their egg-laying rates were still higher while roaming. *tbh-1* mutants defective in octopamine synthesis showed a mild increase in their roaming velocity, but no deficit in egg-laying. Overall, these mutants point to important roles for neuromodulation in regulating behavioral states in *C. elegans*, but do not provide insights into the coupling of egg-laying with the roaming state.

In contrast, dopamine-deficient *cat-2* mutants displayed a striking loss of coupling between egg-laying and the roaming state. Although these animals displayed robust roaming states, their egg-laying rate was dramatically reduced during roaming (*Figure 3A*). In contrast, their egg-laying rate during the dwelling state was unaltered. *cat-2* animals travel at a higher velocity than wild-type animals while roaming (*Figure 3A*), but still display the same postures (*Figure 3—figure supplement 1A*), indicating that they are not broadly defective in the locomotor components of roaming (*Sawin et al., 2000*). Because they have unaltered egg-laying rates during dwelling, which is the more prevalent state, there is only a modest reduction in egg-laying overall in these mutant animals (*Figure 3C*). This suggests that they are not broadly defective in egg-laying, a finding that is consistent with the fact that they were not recovered as egg-laying defective (*egl*) in forward genetic screens. To ensure that this motor coupling phenotype was caused by the loss of the *cat-2* gene, we examined a second, independent null allele of *cat-2* and found that it caused a similar phenotype (*Figure 3C–D*; *Figure 3—figure supplement 1B*). In addition, we rescued *cat-2* expression in the mutant via a transgene and found that this restored higher egg-laying rates during roaming (*Figure 3—figure supplement 1C*). These data suggest that dopamine is necessary for proper coupling of egg-laying to the roaming state and highlights the value of simultaneously quantifying multiple ongoing motor programs within the animal.

We considered whether the reduced egg-laying rates of *cat-2* mutants during roaming, but not dwelling, might be due to a floor effect, where the egg-laying rates during dwelling cannot be further reduced. To test for potential floor effects, we recorded mutant animals with deficits in the egg-laying motor circuit (*egl-1*). These animals showed a dramatic reduction in egg-laying during both roaming and dwelling states (*Figure 3—figure supplement 1D*), suggesting that floor effects do not account for the *cat-2* phenotype.

To further characterize the dopaminergic pathway that couples egg-laying and locomotion during roaming, we examined mutants lacking each of the seven known dopamine receptors in *C. elegans* (*Figure 3E*; *Figure 3—figure supplement 1E*; *Figure 3—figure supplement 2*). None of these single mutants showed any egg-laying phenotypes. However, we found that mutant animals lacking the two D2-like dopamine receptors, *dop-2* and *dop-3*, displayed reduced egg-laying rates during roaming, but not dwelling, closely matching the *cat-2* mutant phenotype. These results suggest that *dop-2* and *dop-3* act together to regulate egg-laying during the roaming state.

In analyzing these mutant datasets, we considered whether the egg-laying phenotypes of the dopamine mutants could be due to an indirect effect, where increased forward velocity during roaming in dopamine pathway mutants reduces egg-laying. However, an analysis of velocity and egg-laying rates across the receptor mutants suggests that this is not the case: *dop-2;dop-3* double

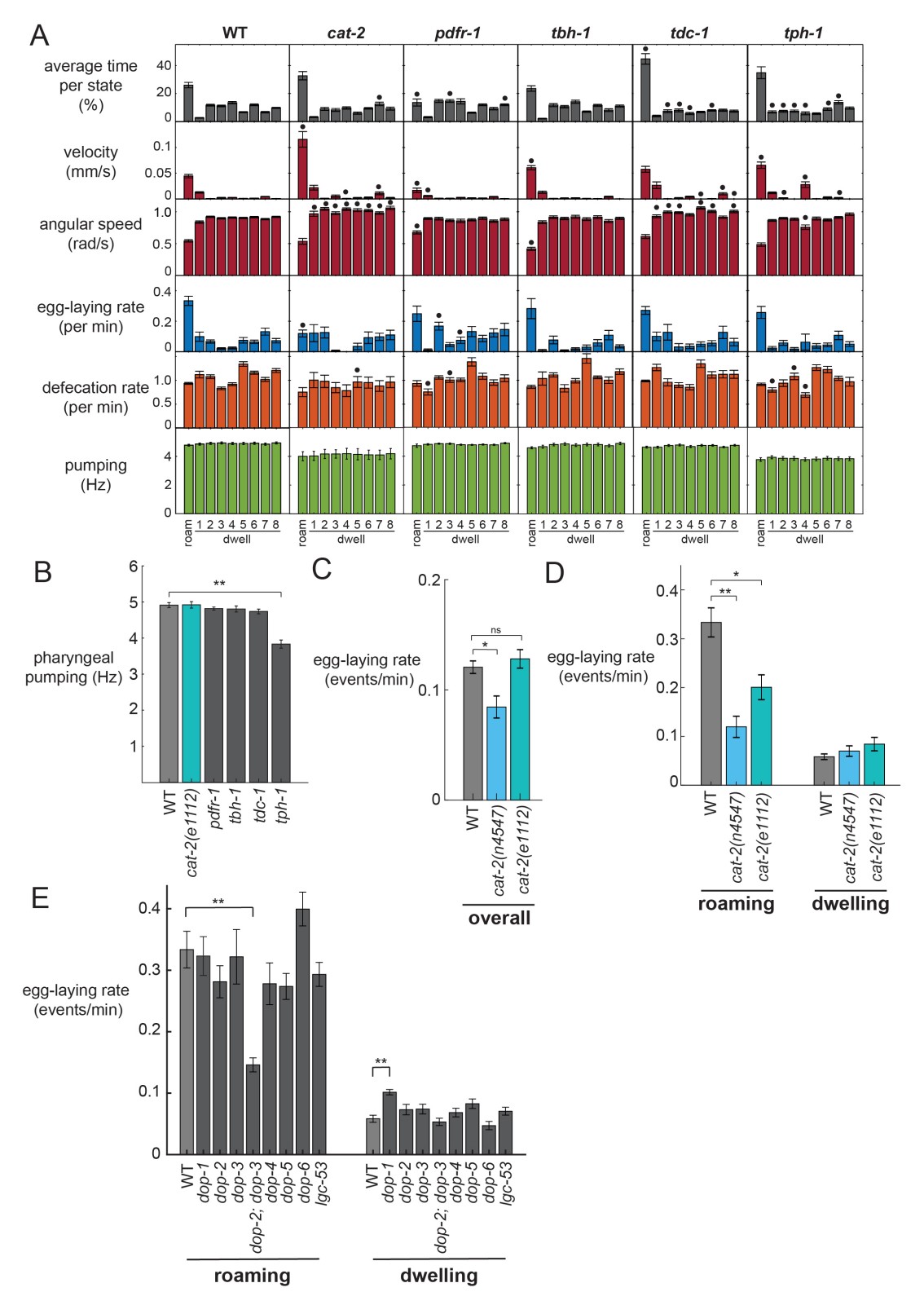

**Figure 3.** Analysis of neuromodulation mutants reveals a role for dopamine in state-dependent egg-laying. (A) Behavioral parameters of animals of the indicated genotypes across the posture-HMM states. Black dots indicate significant changes; *p<0.05, Bonferroni-corrected Mann-Whitney U test. The cat-2 mutant displayed here is the *n4547* allele. (B) On food pharyngeal pumping rates in the neuromodulation mutants (data are pooled across all states). **p<0.001, Bonferroni-corrected Mann-Whitney U test. For (A–B), n = 10–30 animals per genotype. (C) Overall egg-laying rates in the *cat-2* null

*Figure 3 continued on next page*

*Figure 3 continued*

mutants. *p<0.01, Mann-Whitney U test. (D) Egg-laying rates during roaming and dwelling states in two different dopamine-deficient *cat-2* null mutant strains. *p<0.05, Mann-Whitney U test. **p<0.001, Mann-Whitney U test. For (C–D), n = 10–30 animals per genotype. (E) Egg-laying rates during roaming and dwelling for mutant animals lacking specific dopamine receptors. **p<0.01, Bonferroni-corrected Mann-Whitney U test. n = 9–30 animals. All data are shown as means ± SEM.

The online version of this article includes the following figure supplement(s) for figure 3:

**Figure supplement 1.** Additional behavioral analysis of mutant strains.

**Figure supplement 2.** Full behavioral parameters of the dopamine receptor mutant animals.

mutants display normal velocities during roaming (*Figure 3—figure supplement 1E*), but have dramatically reduced egg-laying rates during roaming (*Figure 3E*). This interpretation that the egg-laying effects are not due to locomotion changes is further corroborated by our optogenetics findings below. Altogether, these mutant analyses suggest that dopamine acts through the D2-like receptors DOP-2 and DOP-3 to promote egg-laying during the roaming state.

## The coupling between egg-laying and roaming states enhances the dispersal of eggs across a food source

Animals might coordinate their locomotion and egg-laying in order to lay their eggs in locations that are advantageous for the survival of their progeny. For example, it might be adaptive for animals to broadly disperse their eggs across beneficial environments. We considered how the increased frequency of egg-laying during the roaming state might impact the spatial distribution in which eggs are laid as animals explore a food source. Specifically, we asked whether eggs laid during the roaming state are more dispersed throughout the environment than eggs laid during dwelling, and whether deficits in dopamine signaling reduce the dispersal of eggs. First, to examine the proximity of eggs to one another, we measured the average distance between each egg and its k nearest neighbors (varying the number of k from 1 to 10). As expected, we found that eggs laid during roaming were farther apart from other eggs compared to those laid during dwelling (*Figure 4A*). To gain a more complete view of how animals distributed their eggs, we next measured the extent to

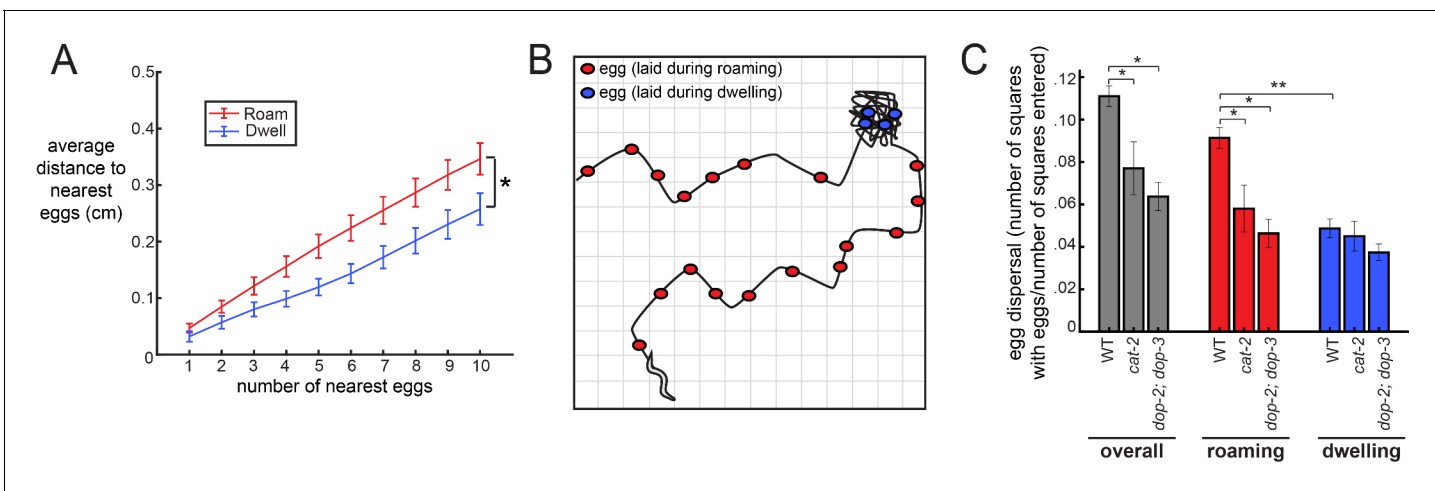

**Figure 4.** The coupling between egg-laying and roaming states leads to greater dispersal of eggs along a food source. (A) Average distance between a given egg and its k nearest eggs for wild-type animals, separated out for eggs laid while roaming and dwelling. *p<0.01, roaming versus dwelling for all k > 3, Bonferroni-corrected Wilcoxon ranked sign test. n = 30 animals. Data are means ± SEM. (B) Schematic of the assay for egg dispersal used in panel C. A grid is virtually superimposed upon the animal's movement path and the number of squares that contain at least one egg is counted and divided by the number of squares where the animal explored. (C) Egg dispersal overall, as well as during roaming and dwelling, for WT (n = 30), *cat-2* (n4547) (n = 10), and *dop-2;dop-3* (n = 19), quantified as the fraction of regions explored with egg-laying events. See panel B for a schematic of the assay. Data are means ± SEM. **p<0.0001, Wilcoxon signed rank test. *p<0.05, Mann-Whitney U test.

The online version of this article includes the following figure supplement(s) for figure 4:

**Figure supplement 1.** Fraction of eggs laid per state in dopamine pathway mutants.

which animals dispersed their eggs across the entire area that they explored. This was quantified by virtually superimposing a grid on the animal's movement path and counting the number of the squares where at least one egg was laid (normalized to the number of squares where the animal traveled; *Figure 4B*). This analysis also revealed a greater dispersal of eggs during roaming (*Figure 4C*). Moreover, dopamine-defective mutants (both *cat-2* and *dop-2;dop-3*) displayed a significant reduction in egg dispersal (*Figure 4C*). Consistent with these results, *cat-2* and *dop-2; dop-3* mutants both laid a higher fraction of their eggs during dwelling, as compared to roaming (*Figure 4—figure supplement 1*). These data suggest that the coupling of egg-laying to the roaming state increases the dispersal of eggs across a food source, which could have important implications for the survival of an animal's progeny.

## Acute changes in dopaminergic neuron activity control state-dependent egg-laying

Our genetic analyses are consistent with the possibility that dopamine signaling plays a direct role in elevating egg-laying rates during the roaming state. To examine whether dopamine acutely regulates egg-laying, we performed optogenetic studies to alter the activity of the neurons that release dopamine. To silence dopaminergic neurons, we used the *dat-1* promoter (*Flames and Hobert, 2009*; *Jayanthi et al., 1998*) to drive expression of the GtACR2 anion channel (*Govorunova et al., 2015*) in all four dopaminergic cell types: CEPV, CEPD, ADE, and PDE (no single-neuron promoters have been described for any of these four cell types). Exposing these animals to light caused them to display lower egg-laying rates that persisted until the light was terminated (*Figure 5A*; additional controls in *Figure 5—figure supplement 1A*). We asked whether this effect reflected a reduction in egg-laying during roaming, dwelling, or both states by separately analyzing egg-laying rates during each of the states during the lights-on period (*Figure 5B*). This analysis indicated that the silencing of dopaminergic neurons reduced egg-laying rates while animals were roaming, but only had a mild, non-significant effect during dwelling. We examined the other behaviors automatically quantified by the tracking microscopes but observed no significant effects of dopaminergic neuron silencing on feeding or defecation, and only a small, transient effect on locomotion (*Figure 5—figure supplement 1B–E*). These data corroborate our genetic findings and suggest that dopaminergic neurons function to acutely promote egg-laying in adult roaming animals.

To examine whether exogenously increasing dopaminergic activity levels is sufficient to drive egg-laying, we acutely activated the four dopaminergic cell types. For these experiments, we used the *dat-1* promoter to drive expression of CoChR, a light-gated cation channel (*Klapoetke et al., 2014*). Activation of *dat-1::CoChR* for three minute durations caused animals to display elevated egg-laying rates (*Figure 5C*; minimal effects on velocity, *Figure 5—figure supplement 1B and F–H*; additional controls in *Figure 5—figure supplement 1I*). This increase in the rate of egg-laying persisted throughout the period of light exposure and was not necessarily time-locked to the moment of light onset. By contrast, acute activation of the egg-laying motor neuron HSN is known to trigger egg-laying events within seconds of light exposure (*Emtage et al., 2012*; *Leifer et al., 2011*). These data suggest that increased dopaminergic neuron activity increases the frequency of egg-laying events.

The observation that dopamine is necessary for proper egg-laying rates during roaming, but not dwelling, might be explained by elevated dopamine release during roaming. Alternatively, dopaminergic activity might be similar across states, but downstream circuits might integrate their detection of dopamine with a locomotion state variable, such that dopamine only enhances egg-laying during roaming. To begin to distinguish between these possibilities, we asked whether the ectopic activation of dopaminergic neurons via optogenetics could increase egg-laying during roaming, dwelling, or both states. We found that light exposure to *dat-1::CoChR* animals elevated egg-laying during the dwelling state, but not during roaming (*Figure 5D*). The finding that the exogenous activation of dopaminergic neurons during dwelling is sufficient to increase egg-laying argues against a model where downstream circuits only respond to dopamine during roaming, but not dwelling. The finding that *dat-1::CoChR* activation does not further enhance egg-laying during roaming may reflect an occlusion effect, perhaps due to the already high egg-laying rates during roaming.

We again considered whether these effects of dopamine on egg-laying could be explained by an indirect effect on locomotion. We analyzed the velocity changes induced by optogenetic dopaminergic neuron silencing and activation. These two manipulations, which caused opposite effects on egg-

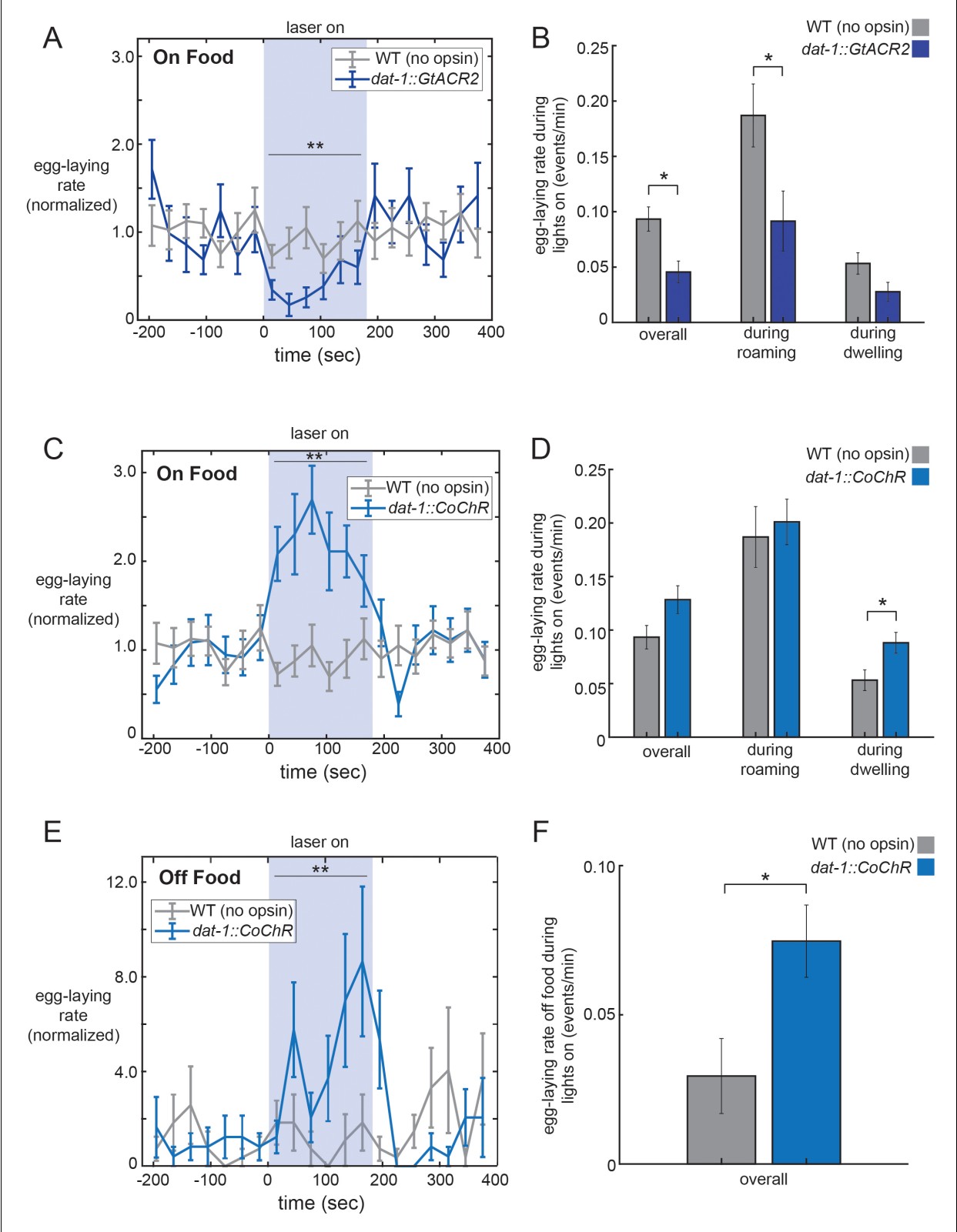

**Figure 5.** Optogenetic control of dopaminergic neurons alters egg-laying. (**A**) Inhibition of dopaminergic neurons via activation of *dat-1::GtACR2* reduces egg-laying rates. Data are shown as egg-laying rates, normalized to pre-stimulation baseline rates. Gray lines show wild-type animals stimulated with the same light pattern. **\*\*p<0.0001, Wilcoxon signed rank test. (**B**) Average egg-laying rates during light exposure for *dat-1::GtACR2* and wild-type animals. Data are shown as overall rates, as well as rates during roaming and dwelling states specifically. \*p<0.05, Mann-Whitney U test. *Figure 5 continued on next page*

Figure 5 continued

For (**A–B**), n = 416 light stimulation events across 12 animals for *dat-1::GtACR2*, and n = 769 light events across 22 animals for wild-type. (**C–D**) Activation of dopaminergic neurons via activation of *dat-1::CoChR* increases egg-laying rates. Data are shown as in (**A–B**). **p<0.001, Wilcoxon signed rank test. *p<0.05 Mann-Whitney U test. n = 1221 light stimulation events across 35 animals for *dat-1::CoChR*, and n = 769 light events across 22 animals for wild-type. (**E–F**) Effects of *dat-1::CoChR* activation in animals that are in the absence of food. Data are shown as in (**A–B**), except only overall egg-laying rates are shown since animals do not roam/dwell in the absence of food. *p<0.05, Mann-Whitney U test. **p<0.0001, Wilcoxon signed rank test. n = 387 light stimulation events across 44 animals for *dat-1::CoChR*, and n = 327 light events across 38 animals for wild-type. Data are means ± SEM.

The online version of this article includes the following figure supplement(s) for figure 5:

**Figure supplement 1.** Additional optogenetic studies and analysis.

laying, caused the same velocity change: a transient decrease in speed at light onset and a transient increase in speed at light offset (*Figure 5—figure supplement 1A*). Therefore, the effects of these opposing optogenetic manipulations on egg-laying cannot be plausibly explained as an indirect consequence of altering locomotion.

Animals have dramatically lower egg-laying rates when they are removed from their food source. Because dopamine signaling is thought to be elevated in the presence of food (*Sawin et al., 2000*; *Tanimoto et al., 2016*), we asked whether increasing dopaminergic neuron activity in the absence of food was sufficient to increase egg-laying rates. Strikingly, we found that activation of *dat-1::CoChR* could drive animals to lay eggs in the absence of food at significantly higher rates (*Figure 5E–F*; velocity in *Figure 5—figure supplement 1J*). Altogether, these data suggest that native dopaminergic neuron activity in adult animals increases the probability of egg-laying during roaming states, and that exogenous dopaminergic neuron activity during dwelling or in the absence of food is sufficient to enhance egg-laying rates.

## Calcium dynamics in dopaminergic PDE neurons are phase-locked to egg-laying during roaming states

The above data suggest that dopamine release can enhance egg-laying and that native dopaminergic signaling in wild-type animals primarily promotes egg-laying during the roaming state. One possible explanation for these observations is that dopamine release might predominate during high-speed roaming. Thus, we next examined the native activity patterns of dopaminergic neurons in freely-moving animals. We constructed a transgenic strain expressing GCaMP6m under the *dat-1* promoter and recorded GCaMP signals in freely-moving animals using widefield imaging, as has been previously described (*Flavell et al., 2013*; *Rhoades et al., 2019*). The CEPD, CEPV, and ADE classes of neurons appeared to have static calcium levels while animals explored food and did not display dynamics that were dependent on locomotion speed (*Figure 6—figure supplement 1A*). However, the dopaminergic PDE neurons displayed robust calcium dynamics as animals freely explored a food lawn (*Figure 6A–B*). PDE neurons have short ciliated sensory dendrites that protrude through the cuticle along the dorsal side of the animal in the posterior section of the body. The PDE axon travels along the ventral nerve cord, from the posterior to the anterior end of the animal. Notably, PDE is the only dopaminergic neuron whose neurite passes in close proximity to the egg-laying circuit and vulval muscles.

To quantify how PDE activity changes during locomotion, we first quantified the extent to which PDE dynamics depend on locomotion. We found that fluctuations in PDE calcium levels were significantly increased during high-speed movement (*Figure 6A*), particularly during roaming (*Figure 6—figure supplement 1B*), a relationship that resembles how animal movement is coupled to egg-laying (*Figure 2F*). As animals moved, PDE activity was highly correlated with the animal's body curvature. PDE was most active when the dorsal body wall muscles were contracted on the posterior end of the body (*Figure 6B* shows an example; *Figure 6C* shows correlations). This correlation between posture and PDE activity was present during roaming and dwelling (*Figure 6C*; *Figure 6—figure supplement 1C*). However, when animals are roaming, repeating waves of muscle contractions pass through the body from head to tail. Thus, one result of the correlation between PDE activity and body curvature was that PDE activity oscillated during roaming and was significantly elevated during a specific phase of each sinusoidal propagating wave (*Figure 6D* shows an example; *Figure 6E* shows averages). Because PDE has an exposed sensory ending, we examined whether this activity

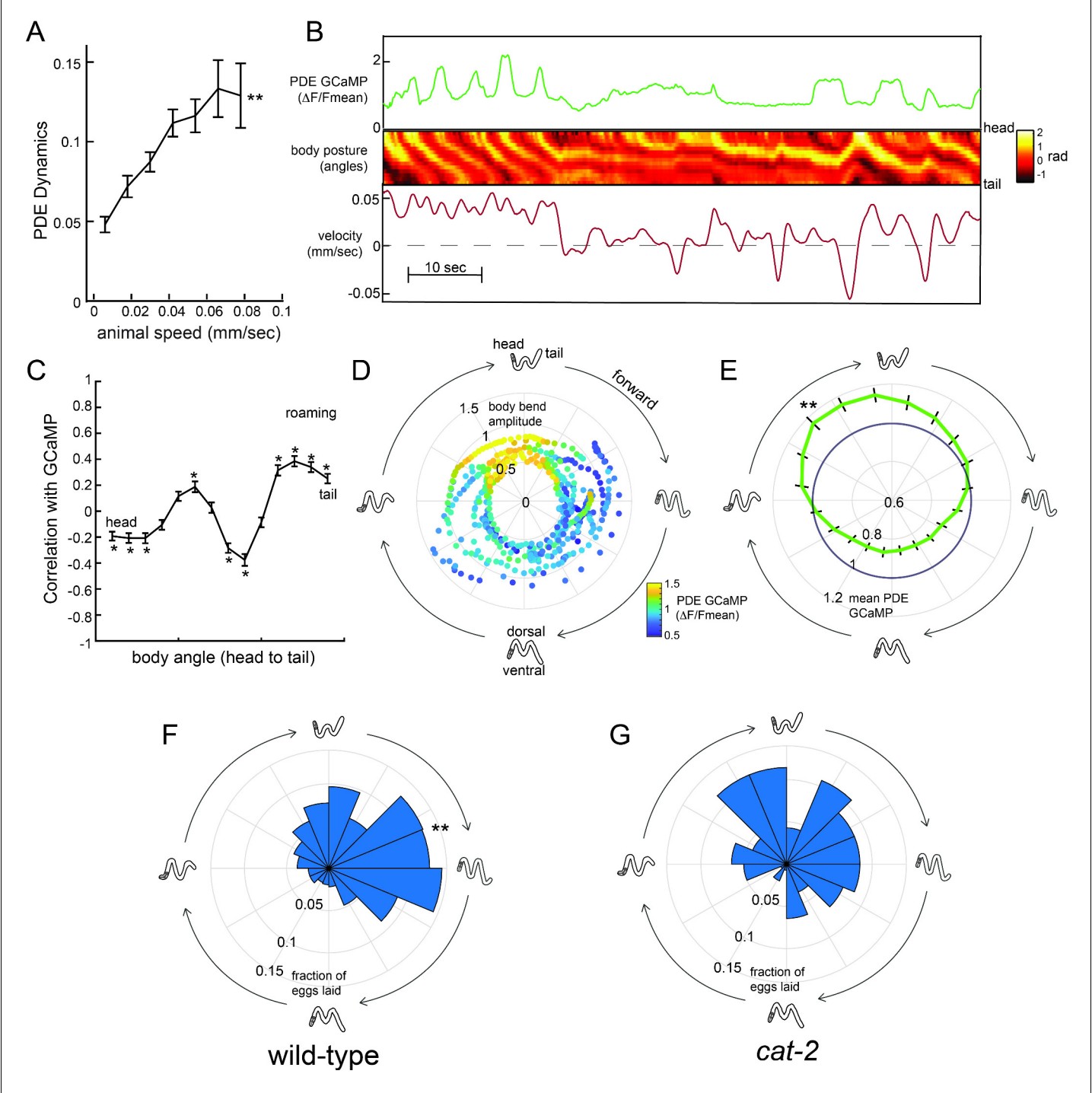

**Figure 6.** Dopaminergic PDE neurons display activity patterns phase-locked to egg-laying during roaming. (A) PDE dynamics increase with animal speed. PDE dynamics here is defined as the absolute value of the time derivative of the PDE GCaMP signal. **p<0.001, empirical bootstrap test. (B) Example dataset from a wild-type animal, showing PDE::GCaMP6m signal, body posture (shown as body angles, from head to tail), and animal velocity. (C) Correlation coefficient of PDE activity with each of the 14 body angles. *p<0.05, empirical bootstrap test (Bonferroni-corrected). (D) Example dataset showing how PDE activity (indicated by color) changes as animals proceed through stereotyped forward propagating bends during roaming. Theta values on the polar plot correspond to the phase of the forward propagating bend; radius corresponds to the depth of the body bends (which was quantified as the standard deviation of the mean-subtracted body angles). Corresponding body postures and the direction of the trajectories during forward movement are indicated. Note that PDE activity reliably increases during a specific phase of the forward propagating bend. (E) Average PDE activity at different phases of the forward propagating bend. Theta values are defined as in (D) and the radius indicates the mean PDE GCaMP signal. **p<0.001, Mann-Whitney U test. (F) Histogram depicting frequency of egg-laying events for wild-type animals at different phases during the forward-

*Figure 6 continued on next page*

*Figure 6 continued*

propagating bend. **p<0.001, Rayleigh z test. (**G**) Histogram depicting frequency of egg-laying events for *cat-2* mutants at different phases during the forward-propagating bend. *p<0.05 versus wild-type, Fisher's exact test. For (**A**), (**C**), and (**E**), n = 39 animals and data are shown as means ± SEM. For (**F**) and (**G**), n = 30 and 10 animals, respectively.

The online version of this article includes the following figure supplement(s) for figure 6:

**Figure supplement 1.** Additional analyses related to PDE activity and egg-laying.

pattern was impacted by the presence of bacterial food in the environment. Indeed, we found that PDE displayed significantly reduced dynamics during forward locomotion in the absence of food (*Figure 6—figure supplement 1D*). These relationships between PDE fluorescence and body posture were not observed when imaging PDE::GFP (instead of GCaMP), indicating that they were not due to motion artifacts (*Figure 6—figure supplement 1E–F*). These data indicate that PDE calcium levels oscillate during forward movement on food, with a reliable activity peak during a specific phase of the forward propagating wave during roaming.

Previous studies suggested that egg-laying events also depend on body curvature (*Collins et al., 2016*), so we also quantified the frequency of egg-laying events during each sinusoidal propagating wave of muscle contractions during roaming. Indeed, egg-laying events were strongly biased to a specific phase of the sinusoidal wave (*Figure 6F*). Notably, the phase of maximal PDE activity overlapped with and shortly preceded the phase of maximal egg-laying. These postural changes that preceded egg-laying during roaming were distinct from those during that preceded egg-laying events during dwelling, even though animals adopted similar postures in both states at the precise moment of egg-laying (*Figure 6—figure supplement 1G*). These data indicate that the animal's gait during roaming leads to a unique coupling of body curvature to egg-laying, in a manner that causes PDE activity to shortly precede the typical body posture for egg-laying.

This relationship raised the possibility that dopamine release during roaming might not only promote egg-laying events overall, but might also bias them to occur during a specific phase of the sinusoidal wave. To test this possibility, we examined the distribution of egg-laying events in dopamine-deficient *cat-2* mutants and found that there was a significant decrease in the phase-dependence of egg-laying (*Figure 6G*). This difference was not due to a general change in body postures during roaming, since wild-type animals and *cat-2* mutants displayed similar postures during forward propagating waves while roaming (*Figure 6—figure supplement 1H–I*). Together, these data suggest that the activity of dopaminergic PDE neurons is phase-locked to egg-laying events during roaming and that dopamine signaling is required for proper coordination of body posture and egg-laying during roaming.

## GABAergic signaling is required for dopamine-induced egg-laying

We next sought to identify the downstream circuits through which dopaminergic neurons act to elevate egg-laying rates. To begin to identify these downstream components, we examined whether specific neurons or neurotransmitters were necessary for dopamine-dependent egg-laying. Egg-laying in *C. elegans* requires contraction of the vulval muscles, which receive many synapses from HSN and VC neurons and a smaller number of synapses from cholinergic VA/VB neurons and GABAergic VD neurons. We optogenetically activated dopaminergic neurons in mutant animals lacking each of these four inputs: (1) *egl-1* mutants lacking HSNs, (2) *lin-39* mutants lacking VCs, (3) *acr-2* mutants with reduced cholinergic neuron activity, and (4) *unc-25* mutants with abolished GABAergic transmission (*Figure 7A* shows fold-change in egg-laying during lights-on, compared to lights-off). *dat-1:: CoChR* activation still elevated egg-laying rates in *acr-2* mutants, suggesting that cholinergic transmission in ventral cord neurons is not essential for these effects. However, *dat-1::CoChR* activation had a reduced effect in *egl-1* and *lin-39* mutants and failed to elevate egg-laying rates in *unc-25* mutants. The finding that HSN and VCs are required for dopamine-induced egg-laying is not surprising, since these motor neurons are centrally involved in driving egg-laying, but the role of GABAergic signaling in egg-laying has not been well-studied, so we examined this interaction more closely.

In *C. elegans*, GABA is produced by a small set of neurons in the head and tail, and by the VD/ DD motor neurons in the ventral cord. To clarify which neurons mediate dopamine-dependent egg-laying, we activated *dat-1::CoChR* in *unc-30* mutants, which lack GABA in the VD/DD neurons, but

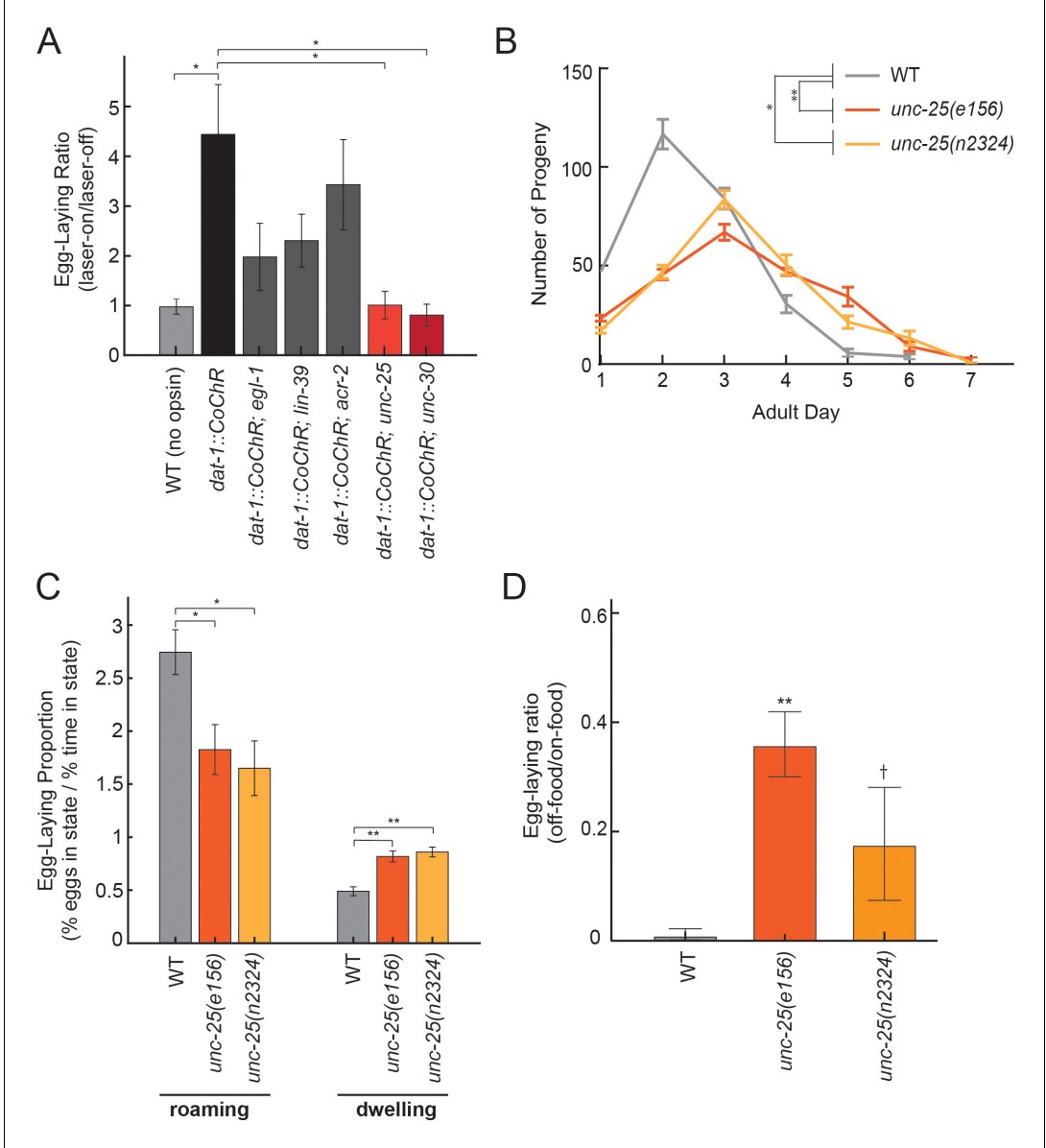

**Figure 7.** Dopamine elevates egg-laying in a GABA-dependent manner. (**A**) Effects of *dat-1::CoChR* activation in genetic mutants with disrupted components of egg-laying circuitry. Data are shown as the fold increase in egg-laying during lights-on period, compared to lights-off period (a value of 1 indicates no effect of the light on egg-laying). *p<0.05, Mann-Whitney U test. n = 9–35 animals per genotype. (**B**) Two independent null mutants lacking *unc-25*, which is required for GABA synthesis, have an altered profile of egg-laying. Data are shown as average eggs laid per day, with the first day being the first 24 hr after L4. *p<0.05 and **p<0.01, effect of genotype in two-factor ANOVA and post-hoc pairwise Dunnett test. n = 5–10 animals per genotype. (**C**) Percent of eggs laid during roaming and dwelling for two independent null mutants of *unc-25*, which are defective in GABA synthesis. Here, we display data in this format to normalize for the reduced brood size in *unc-25*. *p<0.05, Mann-Whitney U test. **p<0.001, Mann-Whitney U test. n = 11–30 animals per genotype. (**D**) Egg-laying in the absence of food is elevated in *unc-25* mutants. Data are shown as a ratio of eggs laid off food divided by eggs laid on food. **p<0.01 and †p<0.1 in ANOVA and post-hoc pairwise Dunnett test. n = 5–6 plates per genotype, each with 10 animals per plate.

not in the other GABAergic cell types in the head and tail (*Figure 7A*). We found that *dat-1::CoChR* failed to elevate egg-laying in *unc-30* mutants, indicating that GABA production by VD/DD neurons is critical for this effect.

A previous study showed that application of muscimol, a GABA_A receptor agonist, strongly inhibits egg-laying (*Tanis et al., 2009*), but the effects of reducing GABAergic signaling on egg-laying had not been previously examined. Thus, we performed experiments to clarify the role of native

GABAergic signaling in egg-laying. Here, we compared wild-type animals to two independent *unc-25* null mutant strains, which both have attenuated GABA synthesis. Compared to wild-type animals, *unc-25* mutants had a slightly lower brood size (~220 eggs per animal vs. ~280 in wild-type) and laid their eggs over their first 5–6 days of adulthood, versus 3–4 days in wild-type animals (*Figure 7B*). As a result, *unc-25* mutants displayed lower overall egg production during their first day of adulthood, but higher egg production as five- and six-day old adults (*Figure 7B*). We examined state-dependent egg-laying in these animals and found that, after adjusting for their lower egg production, one-day old *unc-25* mutants displayed a relative increase in egg-laying during dwelling, as compared to roaming (*Figure 7C*). To test whether GABA also regulates egg-laying across environmental conditions, we examined egg-laying in the absence of food and found that *unc-25* mutants displayed an elevated egg-laying rate off food, compared to wild-type animals (*Figure 7D*). Altogether, these results indicate that native GABAergic signaling regulates egg-laying, specifically playing an inhibitory role during dwelling and in the absence of food. This inhibitory role is consistent with the previous finding that muscimol inhibits egg-laying (*Tanis et al., 2009*). Given that GABAergic signaling is required for dopamine-dependent egg-laying, one possible explanation for these effects is that dopamine release may inhibit GABAergic neurons that function to inhibit egg-laying.

## Discussion

Behavioral states impact the generation of diverse motor outputs, but the neural mechanisms that allow them to exert these widespread effects are poorly understood. To examine this problem at whole-organism scale, we developed a new method that permits simultaneous, automated quantification of diverse *C. elegans* motor programs over long time scales. Analysis of these near-comprehensive records of animal behavior show that there is extensive coordination between the different motor programs of this animal. By combining this approach with genetics and optogenetics, we uncovered a new role for dopamine in promoting egg-laying in a locomotion state-dependent manner. Our results provide new insights into how the diverse motor programs throughout an organism are coordinated and suggest that neuromodulators like dopamine can couple motor circuits together in a state-dependent manner.

To understand how internal states are represented in the brain, it is critical to obtain a quantitative description of the full repertoire of behaviors that are influenced by a given state. The transparency of *C. elegans* allows us to observe each of the motor actions of this animal, even the movements of internal muscle groups like the pharynx. Thus, by using tracking microscopes and machine vision software, we simultaneously quantified the diverse motor programs of this animal. These datasets revealed widespread coordination among distinct motor programs. For example, roaming and dwelling states that were previously described based on locomotion parameters also show robust differences in other motor programs, like egg-laying. Moreover, dwelling states can be segmented into different sub-modes based on reliable differences in body posture and other motor programs. We note that so far our analyses have been limited to well-fed adult animals exploring a homogeneous *E. coli* food source. The structure of *C. elegans* behavior could be quite different when animals are different ages (*Stern et al., 2017*), exposed to different stimuli, or in different physiological states. For example, quiescence/sleep was not observed under our recording conditions, but this state occurs frequently in other conditions. Several factors likely contribute to the coupling between diverse motor programs, such as: changing internal state variables and recruitment of neuromodulators (*Donnelly et al., 2013*; *Flavell et al., 2013*; *Pirri et al., 2009*; *Raizen et al., 2008*; *Van Buskirk and Sternberg, 2007*), corollary discharges and related motor signals that allow motor circuits to interact (*Gordus et al., 2015*; *Liu et al., 2018*), and proprioceptive or environmental feedback (*Goodman and Sengupta, 2019*; *Hu et al., 2011*; *Li et al., 2006*; *Wen et al., 2012*; *Yeon et al., 2018*). Combining the whole-organism behavioral profiling approach described here with recently-developed whole-brain calcium imaging approaches (*Kato et al., 2015*; *Nguyen et al., 2016*; *Venkatachalam et al., 2016*; *Yemini and Hobert, 2020*) could yield a more complete understanding of which neural mechanisms are at work.

To begin to clarify neural mechanisms that underlie the coordination between motor programs, we investigated a particularly robust form of motor program coupling: the increased frequency of egg-laying during high-speed roaming states. In wild-type animals exposed to a food source, egg-

laying rates are approximately six-fold higher during roaming as compared to dwelling. Based on our analysis of the spatial distribution in which eggs are laid, this coupling allows animals to effectively disperse their eggs throughout a food resource. This might serve as an effective 'bet-hedging' strategy to increase the likelihood that at least some of the progeny develop in a beneficial environment. We found that dopamine signaling was required for motor program coupling: mutants with reduced dopamine levels had lower egg-laying rates during roaming but not dwelling. Previous work has shown that native dopamine signaling in *C. elegans* is involved in driving slow locomotion in response to a bacterial food lawn, an effect called the basal slowing response (*Sawin et al., 2000*). Indeed, we observed increased forward velocity in *cat-2* mutants (*Figure 3A*). However, multiple lines of evidence suggest that the effects of dopamine on egg-laying are separable from those on locomotion: (1) *dop-2; dop-3* dopamine receptor mutants have reduced egg-laying during roaming, but normal roaming velocities (*Figure 3E*; *Figure 3—figure supplement 1E*), and (2) optogenetic silencing and activation of dopaminergic neurons have opposite effects on egg-laying, even though they have the same modest effect on locomotion. We have not yet mapped out where the *dop-2* and *dop-3* receptors function to control egg-laying, though it is intriguing that *dop-3* is known to be expressed in VD/DD neurons (*Chase et al., 2004*) in light of our observation that optogenetic dopaminergic neuron activation fails to elevate egg-laying in GABA-deficient mutants (*Figure 7A*). Further studies will be necessary to determine if dopamine acts directly on GABAergic neurons or signals to a different set of *dop-2-* and *dop-3*-expressing neurons to regulate egg-laying.

How does dopamine influence egg-laying rates during the roaming state, but not during dwelling? One possibility is that dopamine release dynamics differ in roaming compared to dwelling. Alternatively, downstream targets could integrate their detection of dopamine with a locomotion state variable, such that dopamine only enhances egg-laying during roaming. Our results argue against the latter, since elevating dopaminergic neuron activity during dwelling is sufficient to increase egg-laying. Moreover, in vivo calcium imaging revealed increased calcium fluctuations in dopaminergic PDE neurons during high-speed roaming. Specifically, we observed that PDE calcium levels oscillate during forward movement, peaking during a stereotyped phase of the forward propagating bend that overlaps with and shortly precedes the peak phase of egg-laying during roaming. We have not yet determined the underlying mechanism that drives these changes in PDE activity. However, since PDE is a ciliated sensory neuron that expresses mechanoreceptors like *trp-4* (*Kang et al., 2010*; *Li et al., 2011*; *Sawin et al., 2000*), it is possible that PDE might be activated by the animal's own movement or by the increased flow of external food along the body during high-velocity roaming. In favor of the latter possibility, we found that PDE dynamics were reduced in the absence of food. Thus, it is possible that PDE receives environmental feedback that indicates the degree of movement along a food source. This type of environmental feedback is known to occur for other sensory modalities – for example, optic flow provides animals with visual feedback of how they are progressing through a visual scene. Although the dopaminergic ADE, CEPV, and CEPD neurons did not display calcium dynamics correlated to behavioral changes, it is possible that tonic levels of dopamine release from these neurons could also play a functional role in egg-laying.

The egg-laying circuit consists of the HSN command neuron and VC, VB, and VD neurons that innervate the vulval muscles. We mapped out downstream effectors of dopamine by optogenetically activating dopaminergic neurons in mutant backgrounds lacking candidate neurons and neurotransmitters. These experiments showed that HSN and VC neurons are important for dopamine's effects on egg-laying and that GABAergic signaling in VD/DD neurons is required for these effects. Recent studies have shown that GABA receptors are found on a multitude of *C. elegans* neurons that do not receive direct synaptic inputs from GABAergic neurons, including HSN and VCs (*Yemini and Hobert, 2020*). Given that VD/DD neurons have GABA release sites in close physical proximity to the egg-laying circuit, it is possible that GABA acts extrasynaptically to influence egg-laying.

It is widely appreciated that neuromodulation contributes to behavioral state control, but the widespread behavioral changes that occur across states make it challenging to understand each neuromodulatory system's specific contribution. The systematic approach described here reveals that in *C. elegans* dopamine elevates egg-laying rates during the roaming state. Previous work showed that PDF neuropeptides drive high-speed locomotion typical of the state and have a more modest effect on egg-laying. The combined actions of these neuromodulators and others likely give rise to the full set of behavioral parameters that define the roaming state, though the nature of their interactions will require additional studies. Recent studies of mammalian anxiety states have shown that

individual neuromodulators like Tac2 neuropeptides can themselves also function in parallel circuits to control different behaviors (*Zelikowsky et al., 2018*). Studies of fixed action patterns in invertebrates also revealed parallel functions of single neuropeptides in distinct circuits (*Kim et al., 2006*; *Scheller et al., 1982*). The whole-organism behavioral profiling approach that we describe here allows the behavioral changes that differ across states to be more fully characterized, as opposed to being analyzed one behavior at a time. Such a level of understanding will be essential to reveal the neural mechanisms that underlie the generation of these brain-wide states.

# Materials and methods

## Key resources table

| Reagent type (species) or resource | Designation | Source or reference | Identifiers | Additional information |
|---|---|---|---|---|
| Strain, strain background (*E. coli*) | OP50 | CGC | ID_FlavellDatabase:OP50 | OP50 |
| Strain, strain background (*C. elegans*) | MT13113 | CGC | ID_FlavellDatabase: MT13113 | *tdc-1(n3419)* |
| Strain, strain background (*C. elegans*) | MT15620 | CGC | ID_FlavellDatabase: MT15620 | *cat-2(n4547)* |
| Strain, strain background (*C. elegans*) | MT9455 | CGC | ID_FlavellDatabase: MT9455 | *tbh-1(n3247)* |
| Strain, strain background (*C. elegans*) | CX11078 | *Stern et al., 2017* | ID_FlavellDatabase: CX11078 | *cat-2(e1112)* |
| Strain, strain background (*C. elegans*) | CX14295 | CGC | ID_FlavellDatabase: CX14295 | *pdfr-1(ok3425)* |
| Strain, strain background (*C. elegans*) | LX645 | CGC | ID_FlavellDatabase: LX645 | *dop-1(vs100)* |
| Strain, strain background (*C. elegans*) | LX702 | CGC | ID_FlavellDatabase: LX702 | *dop-2(vs105)* |
| Strain, strain background (*C. elegans*) | LX703 | CGC | ID_FlavellDatabase: LX703 | *dop-3 (vs106)* |
| Strain, strain background (*C. elegans*) | LX704 | CGC | ID_FlavellDatabase: LX704 | *dop-2(vs105); dop-3(vs106)* |
| Strain, strain background (*C. elegans*) | SWF261 | this study | ID_FlavellDatabase: SWF261 | *dop-4(ok1321);* backcrossed to N2 8x |
| Strain, strain background (*C. elegans*) | CX13111 | this study | ID_FlavellDatabase: CX13111 | *dop-5(ok568);* backcrossed to N2 3x |
| Strain, strain background (*C. elegans*) | RB1680 | CGC | ID_FlavellDatabase: RB1680 | *dop-6(ok2070)* |
| Strain, strain background (*C. elegans*) | MT1082 | CGC | ID_FlavellDatabase: MT1082 | *egl-1(n487)* |
| Strain, strain background (*C. elegans*) | SWF266 | this study | ID_FlavellDatabase: SWF266 | *lgc-53(n4330);* MT13952 was backcrossed to N2 4x |

*Continued on next page*

*Continued*

| Reagent type (species) or resource | Designation | Source or reference | Identifiers | Additional information |
|---|---|---|---|---|
| Strain, strain background (*C. elegans*) | SWF181 | this study | ID_FlavellDatabase: SWF181 | *cat-2(n4547), flvEx87[cat-2 genomic PCR product, myo-3::mCherry]* |
| Strain, strain background (*C. elegans*) | SWF325 | this study | ID_FlavellDatabase: SWF325 | *flvEx133[dat-1:: GtACR2-t2a-GFP, myo-3::mCherry]* |
| Strain, strain background (*C. elegans*) | SWF141 | this study | ID_FlavellDatabase: SWF141 | *flvEx74[dat-1::CoChR, myo-3::mCherry]* |
| Strain, strain background (*C. elegans*) | SWF207 | this study | ID_FlavellDatabase: SWF207 | *egl-1(n487); flvEx74[dat-1::CoChR, myo-3::mCherry]* |
| Strain, strain background (*C. elegans*) | SWF208 | this study | ID_FlavellDatabase: SWF208 | *lin-39(n709); flvEx74[dat-1::CoChR, myo-3::mCherry]* |
| Strain, strain background (*C. elegans*) | SWF258 | this study | ID_FlavellDatabase: SWF258 | *acr-2(n2595 n2420); flvEx74[dat-1::CoChR, myo-3::mCherry]* |
| Strain, strain background (*C. elegans*) | SWF257 | this study | ID_FlavellDatabase: SWF257 | *unc-25(e156); flvEx74[dat-1::CoChR, myo-3::mCherry]* |
| Strain, strain background (*C. elegans*) | SWF314 | this study | ID_FlavellDatabase: SWF314 | *unc-30(e191); flvEx74[dat-1::CoChR, myo-3::mCherry]* |
| Strain, strain background (*C. elegans*) | SWF331 | this study | ID_FlavellDatabase: SWF331 | *flvEx127[dat-1::GCaMP6m, myo-3::mCherry]* |
| Strain, strain background (*C. elegans*) | BZ555 | CGC | ID_FlavellDatabase: BZ555 | *egIs1 [dat-1::GFP]* |
| Strain, strain background (*C. elegans*) | CX14453 | *Bendesky et al., 2012* | ID_FlavellDatabase: CX14453 | *unc-25(n2324)* |
| Strain, strain background (*C. elegans*) | CX13851 | *Bendesky et al., 2012* | ID_FlavellDatabase: CX13851 | *unc-25(e156)* |
| Recombinant DNA reagent | pYCH1 | this study | ID_FlavellDatabase: pYCH1 | *dat-1::GtACR2-sl2-GFP* |
| Recombinant DNA reagent | pSKY1 | this study | ID_FlavellDatabase: pSKY1 | *dat-1::CoChR* |
| Recombinant DNA reagent | pSKY2 | this study | ID_FlavellDatabase: pSKY2 | *dat-1::GCaMP6m* |
| Software, algorithm | ImageJ | ImageJ (http://imagej.nih.gov/ij/) | RRID:SCR_003070 | Version 1.52 |
| Software, algorithm | GraphPad Prism | GraphPad Prism (graphpad.com) | RRID:SCR_002798 | Version 7.03 |
| Software, algorithm | MATLAB | MathWorks (www.mathworks.com) | RRID:SCR_001622 | Version 2019a |
| Software, algorithm | National Instruments | LabView (www.ni.com/en-us/shop/labview.html) | RRID:SCR_014325 | Version 16.0 |
| Software, algorithm | NIS Elements | Nikon (www.nikoninstruments.com/products/software) | RRID:SCR_014329 | V4.51.01 |
| Software, algorithm | The R Project | R (r-project.org) | RRID:SCR_001905 | v3.6.1 |
| Software, algorithm | R Studio | R Studio (rstudio.com) | RRID:SCR_000432 | v1.2.1335 |

## Growth conditions and handling

Nematode culture was conducted using standard methods (*Brenner, 1974*). Populations were maintained on NGM agar plates supplemented with *E. coli* OP50 bacteria. Wild-type was *C. elegans* Bristol strain N2. For genetic crosses, all genotypes were confirmed using PCR. Transgenic animals were generated by injecting DNA clones plus fluorescent co-injection marker into gonads of young adult hermaphrodites. All assays were conducted at room temperature (~22°C).

## Behavioral recording conditions

Animals were staged approximately 72 hr prior to recording. 10 adult animals were picked to an NGM plate seeded with *E. coli* OP50 and left to lay eggs for approximately one hour. Adult animals were removed, and eggs were allowed to develop into adults at room temperature until recorded. For experiments that did not involve optogenetics, animals were left undisturbed for the entire developmental period. For experiments that included optogenetics, animals were picked as L4s to an NGM plate seeded with a bacterial lawn with 50 µM all-trans-retinal (ATR).

The assay plates were standard 10 cm diameter petri dishes, with low-peptone (0.2 g/L) NGM, seeded with 200 µL OP50. 50 µM ATR was added for recordings involving optogenetic stimulation. Roughly circular lawn shapes were created with a spreader, and plates were left to dry overnight at room temperature. We used low-peptone plates because they limited the extent of growth of the bacterial lawn. Thick bacterial lawns could obscure the worm and/or contain deep tracks in them, which reduced the quality of worm tracking. Thus, thin bacterial lawns were more desirable for these recordings.

For each on-food recording, a single 72 hr old adult animal was picked to a plate. Plates were taped onto the microscope stage open and face-down on top of 3 evenly placed spacers to prevent condensation on the stage. For animals recorded in the absence of food, a copper ring (with ~7 cm inner diameter) made of filter paper dipped in 0.02M copper chloride solution was positioned on the plate to prevent the animal from reaching the plate edge. For these recordings, animals were washed in M9 twice and placed on the no-food plate using a glass pipette. Kim wipes were used to remove any remaining liquid.

Recordings began 10 min after animals were positioned on the assay plate and were approximately six hours long for on-food recordings and 1–2 hr long for off-food recordings. Optogenetic stimulation was performed with a 532 nm laser at an intensity of 150 µW/mm$^2$ for three minutes every ten minutes.

All data were included in analyses, except for a small number of datasets that had very poor quality. The criteria for exclusion, which were uniformly applied, were: (1) if the posture could not be determined in >10% of the frames, or (2) if the head vs tail could not be confidently assigned.

## Tracking microscope

### Overview

Recordings were made on three identical tracking microscopes. These microscopes that generate near-comprehensive records of *C. elegans* behavior were named after all-seeing figures: Santa Claus ('he knows when you've been naughty or nice'), Sauron ('the Great Eye is ever watchful'), and Cassandra ('all that you foretell seems true to us'). The tracking microscopes and accompanying software suite were inspired by and share similarities to previous imaging platforms (*Yemini et al., 2013*). The complete bill of materials and build instructions for the tracking microscope are available in a git repository at https://bitbucket.org/natecermak/openautoscope. The image analysis code in R is also available at https://bitbucket.org/natecermak/wormimageanalysisr (copy archived at https://github.com/elifesciences-publications/wormImageAnalysisR; *Cermak, 2020*).

### Microscope optics

The tracking microscope was built predominantly using optics and an optical cage system from Thorlabs. The basic optical path consists of a 4x NA 0.1 Olympus PLAN objective (Thorlabs #RMS4X) with a 150 mm tube lens (Thorlabs #AC254-150-A-ML), yielding a magnification of 3.33X. The image is collected on a monochrome Chameleon3 camera (FLIR, #CM3-U3-13Y3M), with a 1/2" sensor format and 1280 × 1021 pixels (4.8 µm/px at sensor plane, 1.44 µm/px at object plane). This yields a field of view of 1.84 × 1.47 mm. We illuminated the sample with a 780 nm, 18 mW IR LED (Thorlabs

#LED780E) in a transillumination configuration. The LED was diffused using a ground glass diffuser (Thorlabs #DG10-120), and collimated onto the sample using a f=16 mm aspheric condenser lens (Thorlabs #ACL25416U) placed 16 mm from the diffuser surface.

The microscope path also included a 4.5 mW, 532 nm laser diode module (Thorlabs #CPS532) for optogenetic stimulation. The laser beam (3.5 mm diameter) was combined with the main optical path using a 550 nm dichroic (Thorlabs #DMLP550R), then focused on the back focal point of the objective using a f = 100 mm planoconvex lens (Thorlabs #LA1509-A-ML). This illuminated a circular region roughly 1.6 mm in the sample plane, with a calculated average intensity of 150 $\mu$W/mm$^2$.

## Microscope mechanics

We reasoned that mechanical motion of the stage might perturb the worm, so we designed our tracking microscope to move the optical components rather than the sample. We attached the optical train to a gantry on two motorized linear actuators, enabling translation of the optics in the X-Y plane (perpendicular to the optical axis). However, we moved the sample in the Z-axis for focusing.

We used motorized C-beam linear actuators from OpenBuilds Parts Store, which allow roughly 150 mm of travel distance in each axis. They use an 8 mm-pitch leadscrew to translate rotational into linear motion. We drove these leadscrews with NEMA23 stepper motors (Pololu, #1476). Because these motors have 200 steps per revolution and we drove them with 1/32 microstepping, the single-step resolution was 1.25 $\mu$m in each linear axis.

To control the motors, we used DRV8825 stepper controller ICs. These are common stepper motor driver ICs used in CNC and 3D-printing applications. The DRV8825 ICs were controlled by a Teensy 3.2 microcontroller with custom-written firmware, using the AccelStepper library for smooth stepping control. The Teensy 3.2 firmware communicates with the computer over a high-speed serial connection via USB, and continually broadcasts its current position at 200 Hz. The computer controls movement by sending strings to the Teensy telling it to move to a given absolute coordinate – for example 'X200' means go to position 200 in the X-axis, whereas 'Z-3000' means go to position $-3000$ in the Z-axis. Each axis was programmed to move at a maximum speed of 1000 microsteps/s (1.25 mm/s), with a maximum acceleration of 50,000 microsteps/s$^2$ (62.5 mm/s$^2$) for X and Y, and 5000 microsteps/s$^2$ (6.25 mm/s$^2$) for Z.

## Tracking

Our tracking approach was similar to previous reports (*Stephens et al., 2008*). We wrote a custom LabView program (National Instruments; RRID:SCR_014325) to perform tracking. In this program, we acquired and processed images at ~20 Hz. For each frame, we thresholded the image, dilated and eroded the binary image to smooth the outline and fill small holes, and detected connected components. Connected components outside a given size range (typically 20,000–100,000 pixels) were discarded, and of the remaining components, the one nearest to the center of the frame was selected as the target. We calculated the unweighted centroid of that component, and used its deviation from the center of the frame as the error signal. We used a proportional-integral (PI) controller to ensure the worm stayed centered in the image. Typical P and I gains were 0.3 steps/pixel and 0.1 steps/(pixel*second), respectively. The feedback loop rate was set by the frame rate (20 Hz). While we could track worms equally well at faster frame rates (easily exceeding 40 Hz), we found 20 Hz both provided stable and robust tracking. Additionally, 20 Hz was near optimal for recording because below 20 Hz we were unable to accurately quantify pharyngeal pumping, and sampling faster than 20 Hz provided essentially no additional information about worm behavior.

During tracking, each frame was saved as a jpeg file. To reduce the file size, we did not save the entire frame, but instead cropped the image to the target component (described above), padded by 40 pixels in all directions. Typical frames were on the order of 20 kB after jpeg compression.

Because the agar surface is not perfectly level, the microscope must periodically refocus as the worm moves. To automatically focus, we performed a rapid z-series centered on the current z-plane and calculate the sharpness of the image at each z-position. Sharpness was calculated as the variance of the image after high-pass filtering with a Laplacian filter. The microscope then returned to the z-plane that yielded the sharpest image. The entire autofocus operation took ~2.5 s. Autofocusing was triggered 10 min after the previous autofocus, and every time the worm moved more than 2.5 mm from the position of the last autofocus.

In our experience, tracking was extremely robust and worms never exceeded the boundary of the camera frame under typical conditions. However, there were two cases in which tracking sometimes failed – when worms reached the boundary of the dish and when condensation on the plate lid reduced contrast or overall image brightness. When worms reached the dish boundary, there were often dark regions in the image that the worm might enter, at which point the tracking code would no longer be able to identify the worm's position. Similarly, condensation in the optical path tended to decrease overall brightness and contrast, which in some cases prevented the code from identifying the worm as a dark component on a light background.

## Behavior detection

While gross locomotor behavior could be quantified online, most other metrics were not, due to computational requirements. We developed an offline analysis pipeline in R (RRID:SCR_001905) to 1) calculate additional locomotion-related metrics and also quantify 2) body posture 3) pumping 4) defecation and 5) egg-laying.

### Locomotion

We quantified multiple metrics of locomotion on 1- and 10 s timescales, as these capture different aspects of behavior. We measured speed as the magnitude of the displacement of the worm's center of mass and velocity as displacement projected onto the worm's body axis. We also measured the angular direction of the worm and calculated the angular velocity as change in angular direction over time. Finally, we tracked the distance between the worm's center of mass and the bacterial lawn (equal to 0 if the worm was on the lawn), as well as the distance from the worm's nose to the lawn.

### Body posture

Body posture has been the subject of extensive study in *C. elegans* (*Stephens et al., 2008*; *Yemini et al., 2013*). We used a well-established pipeline to quantify body posture. This pipeline follows the same protocol as the online tracking analysis (above), including thresholding, dilation and erosion, and selecting the connected component representing the worm. That component is then thinned by iteratively applying a 3 × 3 look-up-table based thinning filter. This process yields pixels that are along the worm's centerline. We treat these pixels as a graph, in which bright pixels become nodes and we create edges between all pairs of adjacent (8-connected) pixels. If the graph contains cycles, we mark the frame as 'self-intersecting' and do not analyze it further. 'Self-intersecting' frames are most commonly due to the animal exhibiting an omega bend, or a related body posture. If the graph has branchpoints, we iteratively prune the shortest branches until the graph no longer has branchpoints and is thus a single line. We refer to this as the centerline. We smooth the centerline coordinates (31-pixel running mean) to reduce discretization artifacts, and resample the centerline so that it contains 1001 evenly-spaced points. We calculate angles between all consecutive points, and fit this vector of angles to a 2nd-degree b-spline basis consisting of 14 components. This yields a 14-dimensional vector that accurately and succinctly represents the worm's posture. We record the mean and interpret it as the worm's overall orientation. We mean-center the 14-point vector, which we then interpret as representing deviations from the worm's overall orientation. Because this representation allows straightforward and accurate reconstruction of the worm's centerline, we also use this centerline to calculate curvature along the centerline as the inverse of the local radius of curvature.

### Assignment of head/tail

The initial body posture quantification described above arbitrarily designated one end of the worm as the head in each frame. Thus, between any consecutive two frames, the apparent orientation may have flipped 180 degrees due to different assignment of which end was the head. Such a 180-degree flip cannot occur due to the worm's locomotion, as worms cannot reorient themselves by 180 degrees in 50 ms (our frame period). Our first step towards assigning head and tail was to simply ensure frame-to-frame consistency by detecting these 180-degree flips between consecutive frames, and reassigning head and tail of the second frame. To do so, we iterated through frames and if the mean angle difference between frames $i$ and $i+1$ exceeds $\pi/2$ radians, we reassigned

which end was the head in frame *i+1.* Reassigning the head also results in reversing the centerline angle vector and subtracting π from all the body angles.

This was insufficient to ensure consistency across entire datasets, as some frames did not have body angles assigned due to self-intersection of the centerline (described above). Therefore, this first approach could only ensure consistency within each block of contiguous non-self-intersecting frames. We then iterated through contiguous blocks of non-self-intersecting frames and ensured that each block had a similar average centerline intensity (brightness) profile as the previous block. Specifically, we checked whether the average centerline intensity vector of block *j+one* or its reverse was better correlated to that of block *j*. If the reverse was a better match, then the head was reassigned for all frames in block *j+1.*

At this point, the orientation of the worm was consistent across the entire experiment, but the head and tail could still be incorrect. We manually checked whether the head was correctly labeled in one frame; if it was not, then the head/tail assignment was flipped for every frame in the dataset and body angles were also correspondingly reversed. Notably, at no point in this process did we rely on locomotion to determine which end was the head or tail.

## Pumping

Our general strategy to quantify pumping was to detect the position of the grinder as well as the posterior end of the posterior bulb and then count negative peaks in the distance between them. This approach is similar to *Lee et al., 2017*, but performed on a freely-moving worm instead of a worm immobilized in a microfluidic chip.

Because there were occasional errors in estimating the worm's centerline (typically due to the presence of eggs touching the body, especially the tip of the nose/tail), we first eliminated from the analysis all frames in which the worm's length suddenly increased. We identified these frames as those in which the worm length exceeded the $90^{th}$ percentile of the surrounding 10 min by at least 30 μm.

Within the remaining frames, we identified the grinder and the posterior end of the posterior bulb as dark positions in the intensity (brightness) profile along the worm's length. This profile was obtained by averaging the intensity of 8 perpendicularly-oriented pixels for each point on the centerline. We then used spline interpolation to upsample this profile to a roughly 0.2 μm resolution, and looked for the presence of two local minima in the profile. In particular, we sought two local minima amongst the four darkest local minima in the first quarter of the worm's length that were between 8 and 17 μm apart. If these were found, we calculated the distance between them, otherwise we marked the frame as missing data. We refer to this signal as the grinder distance.

For contiguous stretches for which the grinder distance was defined, we bandpass filtered this signal (2nd-order Butterworth filter, passband 0.2–0.8 normalized frequency) and counted peaks that deviated more than 0.6 μm below 0 (the average grinder position). Pumping rates were only defined during time periods in which the grinder distance was not missing.

## Defecation

We used the stereotyped body contraction, known as the defecation motor program (*Thomas, 1990*), to identify defecation events. We first identified proportional fluctuations in the worm's body length by subtracting a moving median filtered (width 50 s) estimate of the worm's length, then dividing by the value of the median filter. This yielded a signal with roughly zero mean, in which deviations represented the proportion by which the length changed. We then applied a matched filter with a kernel consisting of two Gaussians (width 0.9 s) separated by 3.5 s. This is a rough approximation of the body length profile during a typical defecation event. Peaks in the filtered signal were taken to be defecation events.

## Egg-laying

Our general strategy for detecting egg-laying was to look for sudden increases in the worm's width around the vulval region.

We calculated the worm's body width by measuring the distance from the worm's centerline to the dorsal or ventral edge of the worm, perpendicular to the centerline, at 1001 evenly spaced

points along the centerline. To compress these oversampled vectors, we then fit these to a b-spline basis consisting of 30 components, yielding two 30-point 'half-width' (centerline to edge) vectors.

We developed a heuristic in which candidate egg-laying events are detected by meeting the following criteria:

1. the worm's area does not decrease
2. the worm's length does not change by more than 5%
3. the worm's width in any of segments 15–20 (roughly the midpoint to 66% along the worm's length) on either side increases abruptly within 100 ms by at least six pixels.
4. a filter calculated using the worm's width crosses a threshold. In particular, this filter was designed to maximize the signal from an egg suddenly appearing around the vulva, while minimizing the signal from a worm moving alongside an existing stationary egg.

The latter filter was constructed using the average width of 5 segments anterior to the vulval region, five segments around the vulval region, and five segments posterior to the vulval region. A differencing filter of length 250 ms was applied to the width time series from each 5-segment region to detect arrival of eggs, and then the three regions were summed with weights 1, 2, and 1 respectively.

Because *C. elegans* only lays eggs on its ventral side, our analysis then identified the putative ventral side as the side of the worm with more candidate egg laying events. This criterion proved highly reliable in that it correctly identified the ventral side for every worm we analyzed (verified during manual annotation, below). Candidate egg laying events on the dorsal side were ignored.

These criteria yielded generally high sensitivity but poor specificity, often identifying up to >200 frames containing candidate egg laying events (out of >400,000 frames for a six-hour dataset). We then manually verified these candidate frames, typically identifying 20–70 real egg-laying events. For each event we also manually identified the number of eggs laid, as multiple eggs are occasionally laid simultaneously.

## Lawn boundary
Before each experiment, we manually annotated the position of the bacterial lawn by manually directing the microscope to travel around the perimeter of the lawn and record the path coordinates. This enabled post-hoc determination of when the worm was on the bacterial lawn and if not, how far the worm was from it.

## Posture analysis and HMMs
### Overview
As is described in the main text, we used time-varying changes in *C. elegans* posture to learn behavioral states displayed by adult animals feeding on OP50. This pipeline consisted of several data processing steps, described here.

### Data pre-processing
The input data for these analyses consisted of 14-element data vectors at each time-point that describe the worm's posture as a series of relative body angles, from nose to tail (see 'Behavior Detection' section above). We used the location of egg expulsion along the body to determine the ventral side of the body, and inverted the body angles of animals travelling on their left sides, so that all animals were aligned along the same dorsal-ventral axis.

### Posture compendium
To express the worm's posture in a simple, compressed manner, we constructed a compendium of reference postures that encompass the broad range of postures that *C. elegans* animals display while feeding. This approach has been previously employed by other groups (*Schwarz et al., 2015*). Then, for each posture observed at each moment in time, we determined the closest match in the compendium. To construct a compendium, we sampled 100,000 frames from 24 N2 animals and performed hierarchical clustering. We then cut the dendrogram at different depths (1-200) to build compendiums with different numbers of reference postures. Each time, we averaged together the body angles in each cluster to obtain the prototypical posture for that cluster. We could describe the

degree to which the compendium faithfully captured the worm's posture by measuring how much of the variance of the body angle vector was discarded when transforming the actual posture vector to its most similar reference posture. Then, to determine the optimal size of the posture compendium, we examined the variance explained for compendiums with a wide range of sizes (1-200). This curve (*Figure 2—figure supplement 2A*) clearly plateaus beyond a compendium size of ~70, indicating that adding additional postures to the compendium beyond this point fails to capture a great deal more variance. We identified the size at which at least 75% of the postural variance was explained while adding another posture explained less than 1% of additional variance. This size was 100 postures, and the 100-posture compendium explained approximately 76% of the variance (*Figure 2—figure supplement 2A*).

## Transitions between reference postures

To begin identifying stereotyped postural changes, we examined the probability of each reference posture transitioning to the other reference postures. In this analysis, we ignored self-transitions. This transition matrix was constructed using six animals. We then clustered the rows of this transition matrix using k-means clustering to identify sets of postures with similar transition profiles. We observed a clear, symmetric, block-like structure to the clustered transition matrix, indicating that there are sets of postures that animals tend to transition back and forth between. We performed k-means clustering 500 times for a given number of clusters, while varying the number of clusters from 2 to 10. To determine the optimal number of clusters, we measured the degree to which the block-like structure emerged for a given clustering by calculating the ratio of the average intra-group transition probabilities to the average inter-group transition probabilities. Based on this measurement, the optimal number of clusters was found to be eight, and we selected the clustering with the highest value for further analysis. We refer to these eight clusters of reference postures as 'posture groups'.

## Pre-processing for HMMs

To capture slow time-scale changes in body posture, we attempted to describe the typical postures emitted by *C. elegans* animals over three-second intervals (60 frames). We chose this time frame because we found that over three-second intervals animals typically generated postures from a single posture group, suggesting that this time frame was a good match to the posture group transition rate. Nevertheless, because animals sometimes generated postures from two or three groups within a given 3 s bin, we segmented these three-second intervals by clustering. To perform clustering, we described the posture groups emitted over each three-second interval as an eight-element vector where each entry is the fraction of data points belonging to each posture group. We then clustered these vectors by k-means clustering and used silhouette criterion to determine the optimal number of clusters. The optimal number was found to be eight; there was a clear mapping where each cluster consisted of data vectors primarily belonging to a single posture group. This allowed us to describe each three-second interval with a single value 1–8, based on which cluster it mapped to. We will refer to these cluster values as 'coarsened posture groups' in the section below.

## Training and evaluating HMMs

We trained HMMs on the sequences of coarsened posture groups from six animals. The transition and emission matrices were randomly initialized for each model training. To determine the optimal number of hidden states, we trained models across of a range of hidden states (2-12) and used the Bayesian information criterion (BIC) to compare model likelihoods to one another. The curve of median BIC values demonstrated an optimal value at nine hidden states. We trained 200 nine-state models, each from random initial conditions; half were trained on the same group of 6 animals, while the other half were trained on a separate, non-overlapping group of 6 animals. We then evaluated the similarity of these 200 models by measuring the average Euclidean distance between the rows of the emission matrices, that is how similar the emissions were in each state. We found that the models with the highest likelihoods were essentially identical, despite being trained from different sets of animals and different initial conditions (*Figure 2—figure supplement 2E–F*). This observation suggests that model training reliably converges to the same solution. We arbitrarily selected one of

these extremely similar models for further use. To determine the most likely state path for each animal, we used the Viterbi algorithm.

## Eigenworm analyses

In several figure panels, we used the previously developed 'eigenworm' approach (*Stephens et al., 2008*), where the animal's body posture (a 14-element vector of body angles, from head to tail) is expressed as a smaller number of principal components (PCs), or eigenworms. To perform this analysis, we constructed a matrix consisting of each 14-element posture vector observed across all wild-type animals at all timepoints. As is described above, the posture vectors were mean-subtracted so that they were rotationally invariant. We then performed principal component analysis (PCA) on this matrix and, similar to previous studies, found that the top four PCs together explained the majority (87%) of the variance of the body angles. The relative ordering of the eigenworms (1–4, in order of which explains the most variance) was different in our analysis compared to previous studies. We attribute this to the fact that animals in our analysis were on food and, thus, emitting their postures in a different manner from those off of food, as was the case in previous studies (*Stephens et al., 2008*). In all figures, we number the eigenworms according to the variance explained in our dataset.

## Comparing the posture-HMM to continuous models of postural change during dwelling

To examine whether the posture-HMM provides a better description of dwelling behavior than simpler models where animals continuously change their posture during dwelling (vs. switching between discrete states), we generated posture sequences from three different models and asked which provided a closer match to actual animal data. The posture sequences for the three models were generated in the following manner: We first used the hmmgenerate function in MATLAB (RRID:SCR_001622) to generate a sequence of HMM states and emissions whose length matches the entirety of our wild-type dataset (30 animals; 6 hr each). The model parameters used here were the 9-state model parameters described in *Figure 2*. To allow for a direct comparison between the three alternative models, we lumped all eight dwelling sub-modes together and then segmented out each of the dwelling states in this sequence. Then, for each dwelling state, we generated three different synthetic posture sequences corresponding to the three different models:

1. Posture-HMM: We used the emissions from the hmmgenerate output to determine the sequence of posture groups generated by the 9-state HMM (giving rise to a plausible sequence of posture groups specified by the parameters of the 9-state HMM). To generate sequences of postures (with values ranging 1–100; see *Figure 2B–C*) from these vectors of posture groups, we segmented the vector into consecutive stretches of the same posture group and for each stretch we initialized in a random posture within the group. Then 'animals' transitioned between postures within the same group, according to the pairwise transition rates that were measured from real animals (*Figure 2C*). The time spent in each posture was constrained to match actual animals: for each of the 100 postures, we obtained a distribution of posture durations from actual animals and then randomly sampled from the corresponding distribution when generating the in silico posture sequences. This process was re-iterated all the way through each dwelling state.

2. Continuous model with transition rates determined by empirical pairwise transition rates: For each dwelling state, we initialized the sequence of postures in a random posture. The 'animals' then transitioned between postures according to the pairwise transition rates that were measured from real animals (*Figure 2C*). Just like the other models, the time spent in each posture was constrained to match actual animals: for each of the 100 postures, we obtained a distribution of posture durations from actual animals (same distributions were used for all three models) and then randomly sampled from the corresponding distribution when generating the in silico posture sequences. Note that this model is essentially identical to the posture-HMM model, except it lacks the discrete posture group component.

3. Continuous model with transition rates determined by posture similarity: For each dwelling state, we initialized the sequence of postures in a random posture. The 'animals' then transitioned between postures using a rule where the probability of transitioning from each posture to the others was based on how similar the postures were (as measured by the Euclidean distance between the two posture vectors). As in the other models, the time spent in each posture was constrained to match actual animals: for each of the 100 postures, we obtained a

distribution of posture durations from actual animals (the same distributions were used for all three models) and then randomly sampled from the corresponding distribution when generating the in silico posture sequences. Note that this model is identical to Model #2, except the rule for pairwise transitions between postures is based on posture similarity, rather than empirically measured transition probabilities.

To assess how well the sequence of postures from each model matched actual posture sequences recorded from real animals during dwelling, we used the following metric: We measured the average minimum duration of time between observed postures for each pair of postures. For example, we found that when animals were in posture #59, it took an average of 21 s until they next emitted posture #61, but it took an average of 2.7 min until they next emitted posture #31. This metric takes into account the higher order patterns in which animals actually emit their postures, but does not assume any particular underlying structure. We obtained this measurement for each pair of postures in our real recorded data, as well as the synthetic posture sequences from the three models (all analyses were restricted to dwelling states only). The 'error' of each model was taken to be the square of the difference between this metric for a given model and the real animal data (summed across all pairs of postures; *Figure 2—figure supplement 5B*). We note that we also performed a BIC analysis of the 9-state HMM, as compared to HMMs with fewer states (*Figure 2—figure supplement 2D*), to ensure that increased model performance could not be trivially explained by the more complex models having a higher number of parameters.

### In vivo calcium imaging

In vivo calcium imaging of the four dopaminergic cell types was conducted on one-day-old wild-type animals. Animals were positioned on thin, flat slices of NGM agar with a PDMS outer barrier so that animals could not crawl off the agar. OP50 *E. coli* was seeded uniformly on the agar, except for off-food videos. Animals were permitted to equilibrate on the agar slides for 10 min prior to the beginning of each recording. Animals were then recorded with a 4x/0.2NA Nikon objective and an Andor Zyla 4.2 Plus sCMOS camera. Blue light output to animals was 9–20% output from an X-Cite 120LED system for 10 ms of each exposure. Brightfield images (to collect postural and behavioral information) and fluorescence images were collected in an alternating fashion, made possible by alternating which light source was active during each camera exposure. The total frame rate was 20fps, so that each channel (GCaMP and brightfield) had an effective frame rate of 10fps. GCaMP data and behavioral data were thus collected with a 50 ms lag, though for the purposes of our analyses we disregarded this small time lag. GCaMP fluorescence was tracked using a previously-described ImageJ (RRID:SCR_003070) tracking macro (*Flavell et al., 2013*) and behavioral data were extracted using the software that was developed for the tracking microscope analysis (described above).

### Statistics

The details of all statistical tests carried out in this study can be found in *Supplementary file 1*.

## Acknowledgements

We thank Cori Bargmann and members of the Flavell lab for helpful comments on the manuscript, Joshua Powers and Karen Gao for assistance with data annotation, and Andrew Bahle for help with microscope construction. We thank the Bargmann lab, Horvitz lab, and the *Caenorhabditis* Genetics Center (supported by P40 OD010440) for strains. Y-C.H. acknowledges support from the Picower Fellows program. S.W.F. acknowledges funding from the JPB Foundation, PIIF, PNDRF, the NARSAD Young Investigator Award Program, NIH (R01NS104892) and NSF (IOS 1845663 and DUE 1734870).

## Additional information

### Funding

| Funder | Grant reference number | Author |
| --- | --- | --- |
| National Science Foundation | IOS 1845663 | Steven Flavell |

| National Science Foundation | DUE 1845663 | Steven Flavell |
| National Institutes of Health | NS104892 | Steven Flavell |
| JPB Foundation | PIIF | Steven Flavell |
| JPB Foundation | PNDRF | Steven Flavell |
| Brain and Behavior Research Foundation | NARSAD Young Investigator | Steven Flavell |
| JPB Foundation | Picower Fellows Award | Yung-Chi Huang |

The funders had no role in study design, data collection and interpretation, or the decision to submit the work for publication.

### Author contributions

Nathan Cermak, Stephanie K Yu, Conceptualization, Data curation, Software, Formal analysis, Validation, Investigation, Visualization, Methodology, Writing - review and editing; Rebekah Clark, Conceptualization, Data curation, Software, Formal analysis, Supervision, Validation, Investigation, Visualization, Methodology, Writing - review and editing; Yung-Chi Huang, Conceptualization, Data curation, Formal analysis, Validation, Investigation, Writing - review and editing; Saba N Baskoylu, Conceptualization, Data curation, Software, Formal analysis, Validation, Investigation, Writing - review and editing; Steven W Flavell, Conceptualization, Data curation, Software, Formal analysis, Supervision, Funding acquisition, Validation, Investigation, Visualization, Methodology, Writing - original draft, Project administration, Writing - review and editing

### Author ORCIDs

Nathan Cermak  https://orcid.org/0000-0002-8389-8236
Steven W Flavell  https://orcid.org/0000-0001-9464-1877

### Decision letter and Author response

Decision letter https://doi.org/10.7554/eLife.57093.sa1
Author response https://doi.org/10.7554/eLife.57093.sa2

## Additional files

### Supplementary files

• Supplementary file 1. Details about statistical tests. This excel sheet includes a detailed description of each statistical test carried out in this study.

• Transparent reporting form

### Data availability

Data have been uploaded to Dryad (https://doi.org/10.5061/dryad.t4b8gthzf) and are publicly available.

The following dataset was generated:

| Author(s) | Year | Dataset title | Dataset URL | Database and Identifier |
|---|---|---|---|---|
| Cermak N, Yu SK, Clark R, Huang Y-C, Baskoylu S, Flavell SW | 2020 | Behavioral and GCaMP data | https://doi.org/10.5061/dryad.t4b8gthzf | Dryad Digital Repository, 10.5061/dryad.t4b8gthzf |

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
