## [Decision Letter]

**Acceptance summary:**

This paper presents a new imaging setup and analysis pipeline to comprehensively record different *C. elegans* behaviours like egg-laying and locomotion in unprecedented quantitative details and over long time-scales. Using this approach, Cermak and colleagues present evidence indicating that behavioural diversity during locomotion is generated from a finite number of discrete behavioural states, thereby addressing a fundamental problem in ethology. Moreover, the authors discovered that egg laying rates are particularly coupled to the roaming state via dopaminergic circuits, which enables an optimal dispersal of eggs in the environment.

**Decision letter after peer review:**

Thank you for submitting your article "Whole-organism behavioral profiling reveals a role for dopamine in state-dependent motor program coupling in *C. elegans*" for consideration by *eLife*. Your article has been reviewed by two peer reviewers, one of whom is a member of our Board of Reviewing Editors, and the evaluation has been overseen by Ronald Calabrese as the Senior Editor. The reviewers have opted to remain anonymous.

The reviewers have discussed the reviews with one another and the Reviewing Editor has drafted this decision to help you prepare a revised submission.

As the editors have judged that your manuscript is of interest, but as described below that additional experiments are required before it is published, we would like to draw your attention to changes in our revision policy that we have made in response to COVID-19 (https://elifesciences.org/articles/57162). First, because many researchers have temporarily lost access to the labs, we will give authors as much time as they need to submit revised manuscripts. We are also offering, if you choose, to post the manuscript to bioRxiv (if it is not already there) along with this decision letter and a formal designation that the manuscript is 'in revision at *eLife*'. Please let us know if you would like to pursue this option. (If your work is more suitable for medRxiv, you will need to post the preprint yourself, as the mechanisms for us to do so are still in development.)

Summary:

Cermak et al. describe in this manuscript a new imaging setup that allows for both very detailed and comprehensive behavioral analysis while simultaneously performing calcium imaging or optogenetics in *C. elegans*. The setup potentially could be adapted for other small model organisms. The approach allows to record simultaneously the worm's major behaviors, i.e. locomotion, like egg laying, defecation and feeding, across multiple timescales (from ms to hs). Using an unsupervised learning approach adapted from a previous study they classify *C. elegans* locomotion behavior into the known roaming and dwelling states but in addition claim that dwelling can be subdivided into eight different sub-states. A major new finding is that egg laying behavior occurs more frequently during the roaming state, and subsequent genetics experiments indicate that dopamine is involved in this coupling. To further investigate the dopamine circuits, they optogenetically activate or suppress dopamine neurons and measure the effects on egg laying during roaming vs. other states. Indeed, suppression of dopaminergic neurons reduces egg laying mostly during the roaming state. Activation of the same dopaminergic neurons, however, could not increase the egg-laying rate, which the authors attribute to a ceiling effect. Next, they perform calcium imaging of the dopaminergic neuron PDE and find that its activity is highly correlated with the phase of the worm's undulatory gate. Egg laying occurs mostly after the PDE activity peak, and this coupling of PDE and egg laying is abolished in dopamine mutants. Lastly, they identify that GABA production by VD/DD neurons as an inhibitory signal that reduces egg laying during dwelling and off food roaming.

Essential revisions:

1) I am very enthusiastic about the behavioral profiling approach, which to my knowledge is unprecedented in its ability to quantitatively assess qualitatively different behaviors across different times-scales. This has not been achieved in worms before, neither in other models like rodents or flies. Here, comprehensive analysis of behavior relied so far largely on indirect readouts via image processing pipelines, but cannot directly measure the kinematics of behaviors, like gates etc. See for comparison (Berman, 2014) or (Wiltschko, 2015). Unfortunately, I am less excited about how superficially and uncritical the authors describe the major results from this approach.

1.1.) Building-up on an approach developed by Schwarz, 2015, the authors cluster a posture-transitions matrix, and based on its modular structure fit an HMM, finally proposing 9 behavioral states that underlie these blocks of postural sequences. While I am convinced that this procedure was designed carefully and formally segregates blocks of postural sequences that cluster in a statistically significant manner (Please represent the data in Figure 1—figure supplement 1G, J, and N, in the form of a confusion matrix, and calculate the accuracy), based on the lack of any further validation I am less convinced that these indeed represent distinct behaviors in an ethologically meaningful way. Based on the analysis in Figure 2E and upon inspecting Video 1, the 8 dwelling states seem the same behavior to me. Although they might occupy slightly different ranges on a continuous spectrum of behavioral parameters, like speed, I don't see qualitative different behaviors. We would like to see behavioral classifications evaluated more quantitatively. During dwelling, worms are very slow and by chance can pause in different postures while continuously engaging in foraging movements, called. "head sweeping"; why, for example should "shallow bends; ventral head sweeps" be a qualitative distinct behavior than "shallow bends; dorsal head sweeps" or "shallow bends; head sweeping"? Here, the worms might end pausing more frequently in some overrepresented postures but otherwise by chance, while continuing sweeping their heads. This is a trivial explanation that can lead to significant clustering, but speaking against the claim of distinct behavioral states. The manuscript text is very brief and superficial in describing the various dwelling states and no subsequent evaluation of them is provided. Gallagher, 2013, provided evidence for a continuous spectrum of dwelling-like states, which was compelling. Please provide quantifications indicating that the discretization into 8 dwelling states is a better representation of behavior versus the alternative continuous model.

1.2.) I am puzzled why this sophisticated approach fails in detecting well characterized behavioral states, like forward, reversals and omega turns, with known robust transition dynamics. Also, quiescence is not detected, which was recently shown to occur in adults on food. I wonder where these limitations come from. This should at least be intensively discussed in the manuscript.

2) Figure 1F: the eigenworm approach is not described in the Materials and methods, which should be done in a revised manuscript. Also unclear is what the "most commonly observed postures" are and how were they determined. I assume these are the eigenworms as described by Stephens, but then they should be called like this and labelled. If this is the case, then these eigenworms could not be calculated from the entire dataset as described in the text, but from the 4 bins; otherwise they would not differ. This needs to be clarified and a Materials and methods section has to be added. The 2D histograms should not be called eigenworms but 2D histograms of the projection amplitudes a1-a4 like in Stephens. The authors should make more effort in interpreting these histograms, otherwise a non-expert in the eigenworm approach is completely left alone here.

3) I could not understand the most important aspect of this paper: the relationships among behavioral status, dopamine signaling, and egg-laying. A simple interpretation of Figures 4 and 5 is that, during roaming, speed and body bending rates are high, PDE is more activated, and egg-laying is increased. Conversely, during dwelling, speed and body bending rates are low, PDE is less activated, and egg-laying is not increased. If this is the case, DA is not related to behavioral states; instead, it is a link between physical body movement and egg-laying. This possibility is consistent with a simple interpretation of Figure 3 that *cat-2* cannot increase egg-laying during roaming because the DA level is always low in *cat-2*. Body movements may somehow increase egg-laying during roaming, which is independent of DA signaling. Is phase locking of PDE activity and egg laying specific to roaming states? Please repeat all analyses in entire Figure 5 separating the data into roaming vs. dwelling. It is possible that PDE is simply a food gated proprioceptor monitoring speed (or bends) on food. This interpretation speaks against egg laying frequency explicitly coupled to a behavioral roaming state but simply coupled to speed. This should be tested and better discussed.

4) PDE specific rescue of *cat-2* would support the specific involvement of PDE; in case this is too difficult with available promoters or intersectional expression strategies, at least PDE cell ablation should be performed to show that PDE Is specifically required for the coupling of egg-laying to roaming.

5) The opposite effect of *unc-25* mutation on egg laying during roaming vs. dwelling is interesting but confusing; as well as the off-food effect. I wonder whether GABA is really specifically involved in the coupling of egg-laying to roaming, and whether this fits really into the whole story. Further evidence should be provided that dopaminergic and GABAergic circuits indeed signal to each other. As the authors discuss, *dop-2/3* might act in the GABA neurons, specifically VD/DD. The authors should show cell specific rescue of roaming-egg laying coupling via *dop-2/3* in GABA neurons, this would also address at least partially address the problem that the authors did not test multiple alleles of *dop-2/3*.

6) The ethological relevance of egg-laying coupling to roaming is too briefly discussed. Animals spend most of their time on food dwelling and egg laying is not exclusive to roaming; thus, in total a good proportion of eggs laid is during dwelling. These numbers should be quantified here. Moreover, what is the effect of *cat-2* or *dop-2/3* mutation on the overall dispersal of eggs; the authors should be able to extract the dispersal of eggs from their data and quantify this across the various mutants analyzed here. I think such an analysis could significantly elevate the impact of this study from an ethological viewpoint.

---

## [Author Response]

Essential revisions:1) I am very enthusiastic about the behavioral profiling approach, which to my knowledge is unprecedented in its ability to quantitatively assess qualitatively different behaviors across different times-scales. This has not been achieved in worms before, neither in other models like rodents or flies. Here, comprehensive analysis of behavior relied so far largely on indirect readouts via image processing pipelines, but cannot directly measure the kinematics of behaviors, like gates etc. See for comparison (Berman, 2014) or (Wiltschko, 2015). Unfortunately, I am less excited about how superficially and uncritical the authors describe the major results from this approach.1.1.) Building-up on an approach developed by Schwarz, 2015, the authors cluster a posture-transitions matrix, and based on its modular structure fit an HMM, finally proposing 9 behavioral states that underlie these blocks of postural sequences. While I am convinced that this procedure was designed carefully and formally segregates blocks of postural sequences that cluster in a statistically significant manner (Please represent the data in Figure 1—figure supplement 1G, J, and N, in the form of a confusion matrix, and calculate the accuracy), based on the lack of any further validation I am less convinced that these indeed represent distinct behaviors in an ethologically meaningful way. Based on the analysis in Figure 2E and upon inspecting Video 1, the 8 dwelling states seem the same behavior to me. Although they might occupy slightly different ranges on a continuous spectrum of behavioral parameters, like speed, I don't see qualitative different behaviors. We would like to see behavioral classifications evaluated more quantitatively. During dwelling, worms are very slow and by chance can pause in different postures while continuously engaging in foraging movements, called. "head sweeping"; why, for example should "shallow bends; ventral head sweeps" be a qualitative distinct behavior than "shallow bends; dorsal head sweeps" or "shallow bends; head sweeping"? Here, the worms might end pausing more frequently in some overrepresented postures but otherwise by chance, while continuing sweeping their heads. This is a trivial explanation that can lead to significant clustering, but speaking against the claim of distinct behavioral states. The manuscript text is very brief and superficial in describing the various dwelling states and no subsequent evaluation of them is provided. Gallagher, 2013, provided evidence for a continuous spectrum of dwelling-like states, which was compelling. Please provide quantifications indicating that the discretization into 8 dwelling states is a better representation of behavior versus the alternative continuous model.

We performed new analyses to examine whether discretization into 8 dwelling sub-modes provides a better representation of behavior versus more continuous models. We also tested whether this discretization into different sub-modes was supported by independent analyses of the data.

First, we compared how well the HMM described in our study recapitulates data from actual animals, compared to alternative continuous models (Figure 2—figure supplement 5). Here, we compared how well posture sequences generated by three different models resembled data from actual animals. The three models were:

1) The HMM with 8 dwelling sub-modes described in our study

2) A model where animals transition between the 100 compendium postures in a continuous fashion during dwelling, with transition rates between postures determined by those measured in actual animals (i.e. the transition matrix in Figure 2C)

3) A model where animals transition between the 100 compendium postures in a continuous fashion during dwelling, with transition rates between postures determined by posture similarity

We generated posture sequences from all three models while keeping other parameters identical across the models (time spent in a given posture, length of posture sequences generated, durations of dwelling states, etc; see Materials and methods for details). As a metric of how well the posture sequences generated by each model resembled those from actual animals, we computed the average duration of time between observing each pair of postures, resulting in a 100x100 matrix (e.g. animals in posture #59 took 21sec on average to next emit posture #61, but took 2.7min on average to next emit posture #31). This metric captures the higher order patterns in which animals actually emit their postures, but does not assume any particular underlying structure. We compared values from our synthetic posture sequences to those from actual animals and found that the HMM with 8 dwelling sub-modes provided a significantly greater match to actual animal data (Figure 2—figure supplement 5A-B). This was mostly because the synthetic posture sequences from the continuous models had shorter durations of time between postures than was actually observed in the real data. These data suggest that there is long timescale stereotypy in how animals emit their postures during dwelling and that this stereotypy is better recapitulated by the HMM compared to alternative continuous models. In addition this analysis, we also provide a more discernible image of the transition probabilities between the 8 dwelling sub-modes so that this underlying structure can be more easily examined by readers (Figure 2—figure supplement 4C). (finally, we note that our BIC analysis comparing the 9-state HMM to models with fewer states also corroborates this interpretation; Figure 2—figure supplement 2D)

We also examined whether the discretization of dwelling into different sub-modes was supported by analyses of the other behavioral parameters (i.e. not posture, which was used to train the model). Here, we asked whether the inferred moments of state transitions between the dwelling sub-modes were accompanied by any sudden, reliable changes in other behavioral parameters, as might be expected if they were indeed reliable changepoints in behavior (Figure 2—figure supplement 7). Thus, we examined other behavioral parameters at moments of commonly occurring state transitions (not all types of state transitions actually happen, since the transition probabilities from some states to others are essentially zero). Strikingly, we observed that sudden and transient changes in velocity, pumping, and defecation reliably occurred at most of these state transitions (Figure 2—figure supplement 7). The typical behavioral changes that accompany state transitions were different for different types of state transitions (e.g. Dwell4 to Dwell2). These data suggest that the discrete transitions between dwelling sub-modes are reliable changepoints in behavior. We speculate that some of these behavioral changes (DMP event that involves peptide release from the gut; change in food ingestion) could be causal factors in driving state transitions, though further studies will be necessary to fully understand these relationships.

We also added confusion matrices to Figure 1—figure supplement 2G, J, as requested. We could not report data in this form for 2O, since pumping validation was not conducted on a per-pump basis (instead on a per-20s video basis), so we added statistical analysis to this figure panel.

1.2.) I am puzzled why this sophisticated approach fails in detecting well characterized behavioral states, like forward, reversals and omega turns, with known robust transition dynamics. Also, quiescence is not detected, which was recently shown to occur in adults on food. I wonder where these limitations come from. This should at least be intensively discussed in the manuscript.

Based on this suggestion, we examined our data for instances of quiescence, which we defined as a cessation of locomotion, pumping, and defecation for at least 20sec (a permissive estimate, based on Gallagher et al., 2013; Gonzales et al., 2019; and others). We only found five such events in ~180 hours of wild-type data. We included one such example in the revised manuscript (Figure 2—figure supplement 3B) to indicate the potential use of this tracking system for studying quiescence/sleep. We also explain that quiescence is not robustly observed under our recording conditions, which reveals why it was not uncovered as a reliable state in any of our analyses (subsection “Posture-HMM identifies the roaming state and distinct sub-modes of dwelling”, first paragraph).

With regards to forward vs. reverse movement, dwelling animals constantly transition between these directions, often very rapidly. Indeed, individual posture groups (Figure 2C) contain a mixture of postures typical of forward and reverse movement (new Figure 2—figure supplement 1). Therefore, the HMM defined long time-scale states that each typically include multiple F->R and R->F transitions. We describe the relationship of these timescales in better detail in the revised manuscript, as suggested (subsection “Unsupervised classification of *C. elegans* behavioral states based on postural changes”).

Finally, our analysis of posture did not reliably quantify omega bends, as we were unable to algorithmically resolve the head vs. tail when the animal contacted itself. Thus, time points when animals adopted these postures were excluded from our analyses (since we could not be confident that the precise posture was called correctly). This point (which was previously described only in the Materials and methods section) is now mentioned in the Results section as well.

2) Figure 1F: the eigenworm approach is not described in the Materials and methods, which should be done in a revised manuscript. Also unclear is what the "most commonly observed postures" are and how were they determined. I assume these are the eigenworms as described by Stephens, but then they should be called like this and labelled. If this is the case, then these eigenworms could not be calculated from the entire dataset as described in the text, but from the 4 bins; otherwise they would not differ. This needs to be clarified and a Materials and methods section has to be added. The 2D histograms should not be called eigenworms but 2D histograms of the projection amplitudes a1-a4 like in Stephens. The authors should make more effort in interpreting these histograms, otherwise a non-expert in the eigenworm approach is completely left alone here.

We have made the requested changes. The eigenworm analysis is described in the Materials and methods; the “commonly observed postures” are precisely described in the figure legends (Figure 1 legend); the axes of the 2D histograms have been relabeled as a1-a4 like in Stephens; and the Results section has a more detailed interpretation of these histograms (subsection “The distinct motor programs in *C. elegans* are coordinated with one another”).

3) I could not understand the most important aspect of this paper: the relationships among behavioral status, dopamine signaling, and egg-laying. A simple interpretation of Figures 4 and 5 is that, during roaming, speed and body bending rates are high, PDE is more activated, and egg-laying is increased. Conversely, during dwelling, speed and body bending rates are low, PDE is less activated, and egg-laying is not increased. If this is the case, DA is not related to behavioral states; instead, it is a link between physical body movement and egg-laying. This possibility is consistent with a simple interpretation of Figure 3 that cat-2 cannot increase egg-laying during roaming because the DA level is always low in cat-2. Body movements may somehow increase egg-laying during roaming, which is independent of DA signaling. Is phase locking of PDE activity and egg laying specific to roaming states? Please repeat all analyses in entire Figure 5 separating the data into roaming vs. dwelling. It is possible that PDE is simply a food gated proprioceptor monitoring speed (or bends) on food. This interpretation speaks against egg laying frequency explicitly coupled to a behavioral roaming state but simply coupled to speed. This should be tested and better discussed.

We performed new analyses to disentangle the relationships between speed, behavioral state, PDE activity, and egg-laying. Specifically, we aimed to separate out how behavioral state (roam vs. dwell) and animal movement (velocity) each impact egg-laying frequency and PDE activity. First, we plotted animal velocity vs. egg-laying frequency, separating out data during roaming and dwelling (new Figure 2F). If behavioral state were irrelevant, then these curves (roaming and dwelling) would be overlaid, at least for the range of velocities that can be observed in both states. If velocity were irrelevant, then the egg-laying frequency during roaming would be greater than that during dwelling, with no influence of velocity on egg-laying frequencies within each state. What we observed is clear evidence that speed and state are both critical: egg-laying rates are generally higher at higher animal speeds, but the rates during roaming are universally higher than those during dwelling, even when matched for velocity (Figure 2F). Plots of PDE activity show similar relationships (Figure 6—figure supplement 1B). These analyses are most consistent with the interpretation that PDE and egg-laying events are tied to animal speed on food, but are particularly robustly activated by the animal’s gait during roaming. We discuss this interpretation in the revised manuscript (subsections “Dopamine signaling promotes egg-laying during roaming states” and “Calcium dynamics in dopaminergic PDE neurons are phase-locked to egg-laying during roaming states”).

We also plotted the other data in Figure 5, separating out roaming versus dwelling, as requested (Figure 6—figure supplement 1B-C). However, we wish to note that the polar plots in Figure 6D-G were only plotted for roaming and cannot be constructed for dwelling. This is because obtaining the phase of the propagating sinusoidal wave (essential to generate these plots) is trivial during roaming, but impossible during dwelling, since dwelling animals frequently display non-sinusoidal body shapes (kinked postures, etc). To get at this same issue, though, we plotted the average postures of animals prior to egg-laying events during roaming vs. dwelling as average trajectories through PCA space (i.e. in plots with axes representing the projection amplitudes of the eigenworms). These plots that show that roaming animals proceed through a reliable posture trajectory just before egg-laying, which includes the phase of the sinusoidal wave at which PDE is maximally active (Figure 6—figure supplement 1G). However, dwelling animals’ postures are notably different prior to egg-laying events, even though the postures converge to be essentially the same at the precise moment of egg-laying. These data show that the sequence of postures that precede egg-laying are notably different during roaming vs. dwelling. This suggests that the animal’s gait during roaming phase-locks PDE activity to egg-laying events and that the animal’s gait during dwelling results in a different coupling of body posture to egg-laying. These analyses are included in the revised figures, and we expanded the text to fully unpack these new observations (subsection “Calcium dynamics in dopaminergic PDE neurons are phase-locked to egg-laying during roaming states”).

4) PDE specific rescue of cat-2 would support the specific involvement of PDE; in case this is too difficult with available promoters or intersectional expression strategies, at least PDE cell ablation should be performed to show that PDE Is specifically required for the coupling of egg-laying to roaming.

While we agree that this would be an interesting experiment to perform, we already have several lines of evidence in the paper showing that PDE and three other dopaminergic neurons are functionally required for state-dependent egg-laying (opto activation, opto silencing, and mutant analysis of dopamine-deficient animals). We also performed in vivo calcium imaging of all four neurons and only PDE has an activity profile that is correlated with state-dependent egg-laying. Moreover, only PDE has a neurite that comes into proximity of the egg-laying circuit (the other three neurons are confined to the head). While we cannot rule out a role for the other three neuron classes, these existing data already suggest a specialized function for PDE. We have added a discussion of this point in the revised manuscript (Discussion, fourth paragraph). (In addition, we wish to note that we contacted the *eLife* editorial team to inform them that our research lab remains closed due to Covid-19, and they suggested that we proceed with resubmission, given the circumstances)

5) The opposite effect of unc-25 mutation on egg laying during roaming vs. dwelling is interesting but confusing; as well as the off-food effect. I wonder whether GABA is really specifically involved in the coupling of egg-laying to roaming, and whether this fits really into the whole story. Further evidence should be provided that dopaminergic and GABAergic circuits indeed signal to each other. As the authors discuss, dop-2/3 might act in the GABA neurons, specifically VD/DD. The authors should show cell specific rescue of roaming-egg laying coupling via dop-2/3 in GABA neurons, this would also address at least partially address the problem that the authors did not test multiple alleles of dop-2/3.

Our existing data in the paper provide evidence of an interaction between dopamine and GABA: the effect of dopaminergic neuron activation on egg-laying is abolished in GABA-deficient mutants and in *unc-30* mutants lacking GABA specifically in VD/DD neurons. It is possible that this reflects a direct effect of dopamine on VD/DD neurons. Alternatively, there may be intermediate neurons between the dopaminergic neurons and VD/DD neurons, or some other type of interaction. We agree this this is an interesting topic, but teasing this out would likely require a substantial set of new experiments. We have added a discussion about this point to the revised manuscript (Discussion, fifth paragraph). (In addition, we wish to note that we contacted the *eLife* editorial team to inform them that our research lab remains closed due to Covid-19, and they suggested we proceed with resubmission, given the circumstances)

6) The ethological relevance of egg-laying coupling to roaming is too briefly discussed. Animals spend most of their time on food dwelling and egg laying is not exclusive to roaming; thus, in total a good proportion of eggs laid is during dwelling. These numbers should be quantified here. Moreover, what is the effect of cat-2 or dop-2/3 mutation on the overall dispersal of eggs; the authors should be able to extract the dispersal of eggs from their data and quantify this across the various mutants analyzed here. I think such an analysis could significantly elevate the impact of this study from an ethological viewpoint.

We performed new analyses to examine the spatial distribution in which eggs are laid, aiming to understand how egg-laying during roaming vs. dwelling impacts the overall distribution and how dopamine contributes to these effects. We analyzed the spatial distribution of eggs in two ways. First, to understand the degree to which eggs are dispersed relative to one another, we measured the average distances between eggs (e.g. average distance between a given egg and its k nearest neighbors). As expected, this revealed that eggs laid during roaming are more widely dispersed from other eggs than those laid during dwelling (Figure 4A). Next, to examine how effectively each animals disperses its eggs overall, we quantified the degree to which each recorded animal dispersed its eggs over the entire area that it explored (by virtually superimposing a grid on the plate and quantifying the # of squares with eggs, normalized to area explored). This revealed that eggs were more widely dispersed across the explored area during roaming as compared to dwelling (Figure 4B-C). Notably, this assay also showed that *cat-2* and *dop-2;dop-3* mutants displayed a significant deficit wherein they did not disperse their eggs as widely as wild-type animals (Figure 4B-C). These data suggest that the coupling of egg-laying to roaming increases the degree to which animals disperse their eggs while exploring a food source. This could have important implications for the overall survival of an animal’s progeny, which we discuss in the revised manuscript (subsection “The coupling between egg-laying and roaming states enhances the dispersal of eggs across a food source”; and Discussion, third paragraph).

We also provide plots showing the percent of eggs laid during roaming vs. dwelling for WT animals and dopamine pathways mutants, as requested (Figure 4—figure supplement 1). ~60% of eggs are laid during roaming in WT, but this is significantly reduced in *cat-2* and *dop-2;dop-3* mutants.